# CBF-LLM: Safe Control for LLM Alignment

## Abstract

This paper proposes a control-based framework for aligning large language models (LLMs) by leveraging a control barrier function (CBF) to ensure user-desirable text generation. The presented framework applies the CBF safety filter to the predicted token generated from the baseline LLM, to intervene in the generated text. The safety filter includes two significant advantages: this safety filter is an add-on type, allowing it to be used for alignment purposes without fine-tuning the baseline LLM, and if there is an evaluation model regarding the desired alignment, it can be directly applied to the filter design. The overall text-generation system is implemented with Llama 3 and a BERT model, aiming to generate positive text. Finally, further applications and limitations of the CBF-LLM for other alignment tasks, including topic-keeping and hallucination mitigating, are discussed.

## 1 Introduction

While large language models (LLMs) are known to have strong language understanding and generation abilities, they can also generate harmful, biased, or toxic content (Shen et al., 2023; Minaee et al., 2024). Alignment of LLMs ensures that they generate content that is "desirable" for the user, typically meaning content that is safe and ethical. Various approaches for LLM alignment have been presented (see the works by Shen et al. (2023); Minaee et al. (2024); Wang et al. (2023) and reference therein).

The major approach to the alignment is reinforcement learning from human feedback (RLHF, Ouyang et al. (2022)), where a reward model is constructed by human feedback and used for the training of LLMs. Variants of RLHF architectures are also proposed, such as Safe-RLHF by Dai et al. (2024), SENSEI by Liu et al. (2022), and f-DPG by Go et al. (2023), and their implementations are presented, such as training pre-trained LLMs (Bai et al., 2022a; Zhou et al., 2023), and applications like information-seeking chatbot (Glaese et al., 2022). Collecting human feedback with data is time-consuming and expensive. To overcome the drawback, RL from AI Feedback (RLAIF) is presented instead of using human labels (Bai et al., 2022b). In addition, the method to construct the training data automatically is proposed (Kim et al., 2023). Furthermore, to reduce the computational cost, direct preference optimization (DPO) is proposed (Rafailov et al., 2023), where the training data is directly used for training LLMs without accessing the reward model. A common feature of alignment methods like RLHF and SFT is that they modify LLMs' model parameters.

An alternative approach for LLM alignment is to directly intervene in the input prompt or output of LLMs, rather than modifying the model parameters. In-context learning (ICL, Dong et al. (2024)) is a major approach for intervening in the input prompt. In ICL, a few demonstrations are provided in prompt to instruct the LLMs on the task, including few-shot learning (Brown et al., 2020; Zhao et al., 2024). As the methods for intervening in the output, the work by Cao et al. (2021) proposes a method to format output for retrieval application, and the work by Keskar et al. (2019) proposes a repetition penalty to prevent LLMs from repeating the same words and expressions. In addition, the Transformers module provides some functions to modify the output, such as `NoBadWordsLogitsProcessor` and `MinLengthLogitsProcessor` (HuggingFace).

One can view the intervention approach to alignment, which avoids undesirable output, as analogous to "collision avoidance", the most fundamental problem in control engineering. In control engineering, various studies are conducted for safety assurance, including collision avoidance (Isaly et al., 2024; Dawson et al., 2023; Nishimura & Hoshino, 2024). A promising approach for collision avoidance is the control barrier function (CBF), as studied in theoretical works by Ames et al. (2019); Taylor & Ames (2020); Lopez et al. (2021) and in real-world applications by Hu et al. (2023). Just as a vehicle's trajectory is intervened to avoid collisions, LLM's output can be intervened to prevent undesirable content. This paper draws an analogy between vehicle and LLM, as illustrated in 1.

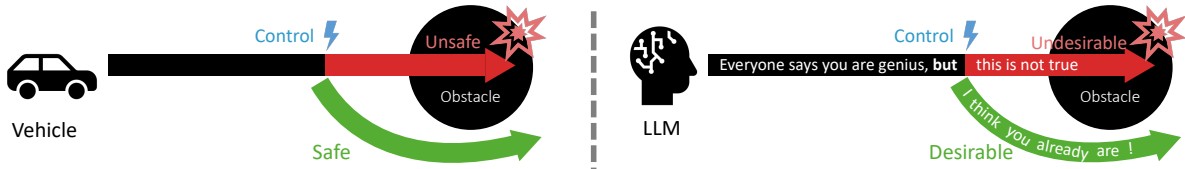

Figure 1: Concept of CBF-LLM. Left: Collision avoidance in a vehicle control system, Right: *Collision avoidance* in text-generation by LLMs. Analogy and differences are discussed in Appendix B.

This paper proposes a framework for control-based LLM alignment by applying a safety filter that intervenes in the LLM output to generate user-desirable outcomes. To this end, we leverage the idea of the CBF to improve the safety and controllability of the outputs of LLMs. We aim to design a CBF-based safety filter that intervenes in the output of LLMs into the user's desired content. The CBF filter and the baseline LLM constitute a novel text generation system, which we call "CBF-LLM". This paper also conducts the text-generation experiment on CBF-LLM. In the experiment, the CBF control enabled the alignment task to be completed with fewer interventions.

CBF-LLM is one of the controlled decoding alignment methods. Controlled decoding alignment has been addressed in the literature, including works such as Mudgal et al. (2024), Zingale & Kalita (2024), Safedecoding (Xu et al., 2024), FUDGE (Yang & Klein, 2021), and DeAL (Huang et al., 2024). The common feature is to intervene in the token probability based on a reward of generated text. On the other hand, our CBF-LLM intervenes in the token probability based on the *change* in reward caused by adding a token, rather than just the generated text. Moreover, CBF-LLM is distinct from previous research in that CBF-LLM considers the constraint of preventing undesirable text from being generated. This paper also attempts to bridge the gap between control engineering and natural language processing. While the *theoretical analysis* of LLMs, such as reachability, has been studied in the works by Bhargava et al. (2024) and Soatto et al. (2023), this paper presents a *design* method of LLM-based text-generation systems.

The other contributions of this paper are as follows: 1. The proposed CBF-LLM is realized in an add-on manner to a baseline LLM: an external filter is simply added to the LLMs without accessing their model parameters. In this sense, CBF-LLM is broadly applicable to various LLMs, as is designed independently of the underlying LLM. 2. CBF-LLM is implemented with Llama 3 and a RoBERTa model in Section 4. The result shows that CBF-LLM is more reliable compared with the previous approaches, in the sense of output alignment.

"Safe Control" addressed in this paper reflects a broader interpretation of "safety". We mean safe control as the ability to regulate LLM output to produce text that not only adheres to ethical standards but also aligns with user-specific objectives.

The rest of this paper is organized as follows. In Section 2, the basic theory of CBF is provided, and the structure of nominal LLM is reviewed. In Section 3, the concept and design of CBF-LLM are proposed. In Section 4, the experiment of CBF-LLM is conducted. Finally, in Section 5, the conclusion of this paper is presented.

Notation: symbol $V[i]$ represents the $i$-th element of the vector $V$.

## 2 PRELIMINARY

### 2.1 CONTROL BARRIER FUNCTION FOR SAFE CONTROL

Control barrier function (CBF), developed in the control community, provides safety assurance in control systems (Ames et al., 2019). This section briefly reviews CBF and CBF-based safety control.

Consider the following dynamical system to be controlled:

$$\dot{x} = g(x, u), \tag{1}$$

where $x \in \mathbb{R}^n$ is the state variable of the object being controlled, and $u \in \mathbb{R}^m$ is the action applied to the object. The function $g$ represents the system dynamics; how this object is affected by the current state $x$ and action $u$. A typical example of the system (1) includes a vehicle dynamics, where the state $x$ is coordinate, velocity, angle, etc, and the action $u$ is accelerator pedal depression, steering angle, etc.

We aim to design the assisted control system with safety assurance. As for safety, we let the safe and unsafe sets be denoted by $\mathcal{S} \subseteq \mathbb{R}^n, \bar{\mathcal{S}} \subseteq \mathbb{R}^n$, respectively. Then, the safety means to constrain $x$ within the safe set $\mathcal{S}$, i.e., $x \in \mathcal{S}$. Consider that the nominal action $u_{\text{nom}} \in \mathbb{R}^m$ is provided which might violate the safety, i.e., the driver $u_{\text{nom}}$ might generate the unsafe state $x \in \bar{\mathcal{S}}$ on a car. Then, we address the problem of designing "safety filter" $F : \mathbb{R}^m \to \mathbb{R}^m$, as follows:

**Problem 1** (Safety Filter). Find the safety filter $F : \mathbb{R}^m \to \mathbb{R}^m$ such that the system (1) with $u = F(u_{\text{nom}})$ generates $x(t) \in \mathcal{S}$ for all nominal actions $u_{\text{nom}}$ for all time $t$.

As a preliminary, we design a continuously differentiable function $h : \mathbb{R}^n \to \mathbb{R}$, called a "constraint function", such that $h(x) \geq 0$ if $x \in \mathcal{S}$, and $h(x) < 0$ if $x \in \bar{\mathcal{S}}$ hold. The safety is equivalent to the constraint: $h(x) \geq 0$. In the assisted driving example, the nominal action $u_{\text{nom}}$ is a manual action by the driver, and the unsafe set $\bar{\mathcal{S}}$ includes the positions of obstacles like pedestrians. The function $h$ can be the distance between the vehicle and the obstacle. With the manual action by the driver $u_{\text{nom}}$, the vehicle may enter the unsafe set, such as colliding with obstacles. The safety filter $F$ needs to modify $u_{\text{nom}}$ to output $u$ such that $h(x) \geq 0$ holds, meaning that the vehicle never collides with obstacles.

To construct the safety filter $F$ that keeps the safety constraint $h(x) \geq 0$, the control barrier function filter (CBF filter, Gurriet et al. (2018); Ames et al. (2019)) is presented. The CBF filter intervenes in the nominal action value $u_{\text{nom}}$ to introduce a safe state of the object by finding $u$ as follows:

$$\min_u \ (u_{\text{nom}} - u)^2, \tag{2}$$

$$\text{s.t. } \dot{h}(x) \geq -\alpha(h(x)), \tag{3}$$

where $\alpha : \mathbb{R} \to \mathbb{R}$ is a class-$\mathcal{K}$ function which holds $\alpha(0) = 0$ and monotonically increasing, i.e., $\frac{d\alpha(x)}{dx} > 0$. In the assisted driving example, when the manual action $u_{\text{nom}}$ is expected to cause a collision with an obstacle, the CBF filter intervenes in $u_{\text{nom}}$ to provide safe driving. To constrain the action $u$ by the CBF filter, the following theorem on the safety assurance holds.

**Theorem 1.** Suppose that the CBF constraint (3) holds for all time. Then, the state $x \in \mathcal{S}$ holds for all time.

**Remark 1.** The objective function (2) ensures that the filtered action $u$ remains as close as possible to the nominal action $u_{\text{nom}}$. In this sense, the CBF filter archives safety by the "minimum" intervention.

The CBF filter is capable of applying in discrete-time systems by re-formulating the CBF constraint (3) as follows (Zeng et al., 2021):

$$\Delta h(k) = h(k+1) - h(k) \geq -\alpha(h(k)), \tag{4}$$

where $k$ is a discrete time.

## 2.2 TEXT GENERATION BY LARGE LANGUAGE MODELS

This section reviews and analyses the text generation by large language models (LLMs) while particularly focusing on their structure. In this paper, "text" means the sequence of tokens and $\mathcal{X}$ denotes the set of the texts. The text corresponding to a specific expression in natural language is displayed as $x = x(\texttt{"<text>"})$. For example, the text $x \in \mathcal{X}$ for "Have a nice day." is displayed as $x(\texttt{"Have a nice day."})$. Each token $t$ is identified by a positive integer, i.e. $t \in \{1, \ldots, N\} =: \mathcal{T}$, and $N$ is the number of tokens the LLM has. The token corresponding to a specific expression in natural language is displayed as a numerical constant $t_{\texttt{<token>}}$. For example, the token for "dog" is displayed as $t_{\text{dog}}$. The function $\texttt{concat} : \mathcal{X} \times \mathcal{T} \to \mathcal{X}$ concatenates text and token to output a text. For example, $\texttt{concat}(x(\texttt{"Have a nice"}), t_{\text{day}}) = x(\texttt{"Have a nice day"})$.

Text generation is performed by iteratively adding a new token, starting from the initial text. Let $x_0 \in \mathcal{X}$ be the initial text, and $k \in \{0, 1, \ldots\}$ be the "time", which counts the number of tokens added during the generation. Then, the text generation from the initial text $x(0) = x_0$ is expressed by a discrete-time dynamical system as

$$\begin{cases} P(k) = G(x(k)), \\ t^*(k) = C(P(k)), \\ x(k+1) = \texttt{concat}(x(k), t^*(k)). \end{cases} \tag{5}$$

In the expression, the symbol $G : \mathcal{X} \to \mathbb{R}^N$ represents the token predictor, which drives the input text $x$ to output the probability vector $P \in \mathbb{R}^N$, where $P[t], t \in \mathcal{T}$ displays how probable that the token $t \in \mathcal{T}$ would

follow by the text $x$. The symbol $C$ is the token selector, which selects the next token $t^*$ which follows $x$ based on the probability vector $P$. The examples of $C$ includes greedy algorithm. Finally, the symbol `concat` is the concatenator, which concatenates the new token $t^*$ with the text $x$ to derive a new text. The symbol $Z^{-1}$ represents the time delay, playing as the memory for the text $x$. The overall structure of the text generation, described by (5), is drawn in Appendix A.

The token predictor $G$ plays a central role in the text generation and is typically realized by LLMs such as GPT-2, and Llama 3. Let the text-generation LLM decoder as $f_{\text{LLM}} : \mathcal{X} \to \mathbb{R}^N$ and $T \geq 0$ be the temperature. Further, recall that $P[t], t \in \mathcal{T}$ display how probable the token $t$ would be followed by the text $x$. Then, the probability $P$ is calculated as follows:

$$P[i] = \text{softmax}(f_{\text{LLM}}(x)[i]/T) = \frac{\exp(f_{\text{LLM}}(x)[i]/T)}{\sum_{j=1}^N \exp(f_{\text{LLM}}(x)[j]/T)}, \tag{6}$$

## 3 CBF-LLM

This section presents the control-based alignment of text-generation systems and their detailed implementation. Symbols and their meanings used in this paper are summarized in Table 1.

Table 1: Symbols and their meanings

| Symbols | Meaning in CBF | Meaning in LLM Alignment (CBF-LLM) |
|---|---|---|
| $x$ | State of the controlled object, $x \in \mathbb{R}^n$. | Generated text, $x \in \mathcal{X}$. |
| $h$ | Constraint function, $h : \mathbb{R}^n \to \mathbb{R}$. | Language-constraint function (L-CF), $h : \mathcal{X} \to \mathbb{R}$. |
| $\mathcal{S}$ | Safe state set. | Desirable text set. |
| $\bar{\mathcal{S}}$ | Unsafe state set. | Undesirable text set. |

The alignment discussed in this paper aims to ensure desirable text generation by weak intervention to the output of LLMs. To clarify the meaning of "desirable", we let the desirable and undesirable text sets be $\mathcal{S} \subseteq \mathcal{X}$ and $\bar{\mathcal{S}} \subseteq \mathcal{X}$, respectively, based on the respective alignment goals.

**Example 1.** Suppose that the alignment goal is set to generate non-toxic content. Then, $\mathcal{S}$ is the set of non-toxic text, and $\bar{\mathcal{S}}$ is the set of toxic texts. For mathematical procedures, we consider $\mathcal{S}$ and $\bar{\mathcal{S}}$ to represent "all" non-toxic and toxic text samples. More examples of $\mathcal{S}$ and $\bar{\mathcal{S}}$ are seen in Section 4.

One simple idea of achieving the alignment goal is to force the text generation to stop when the generated text $x$ turns undesirable, i.e., $x \in \bar{\mathcal{S}}$. However, this method involves a strong intervention in the baseline LLM, which renders the original capabilities of the baseline LLM meaningless. To overcome the drawback, we make the intervention strength adjustable, which enables the text-generation system to achieve the alignment goal with a *weak* intervention.

The presented text-generation system, including an LLM and a safety filter, is constructed based on the CBF described in Subsection 2.1. The overall system is called CBF-LLM and its structure is shown in Fig. 2. CBF-LLM extends the nominal text-generation system shown in (5) by adding the safety filter (yellow box) between the token predictor $G$ and the token selector $C$. The safety filter manipulates the probability $P$ to satisfy the specified alignment task. The safety filter is composed of filter $F$ and normalizer $R$. In the same manner as (5), the blocks of $G$, $C$, `concat`, and $Z^{-1}$ represent the token predictor, token selector, concatenator, and time delay, respectively. The components of CBF-LLM are described in detail as follows.

Recall that the token predictor $G$ mainly implies a generative language model such as LLM. It retrieves the current text $x \in \mathcal{X}$ and outputs the probability of the next token, $P \in \mathbb{R}^N$. For each token $t \in \mathcal{T}$, the probability $P[t]$ indicates how probable each token $t$ is followed after the text $x$.

The defining feature of the CBF-LLM is the presence of filter $F$, which filters $P$ to generate the modified probability $Q \in \mathbb{R}^N$. The filter $F$ is designed to ensure the desirable text generation, i.e., $x \in \mathcal{S}$. To this end, we design the CBF filter in $F$ by using the function $h : \mathcal{X} \to \mathbb{R}$ such satisfying

$$\begin{cases} h(x) \geq 0, & x \in \mathcal{S}, \\ h(x) < 0, & x \in \bar{\mathcal{S}}. \end{cases} \tag{7}$$

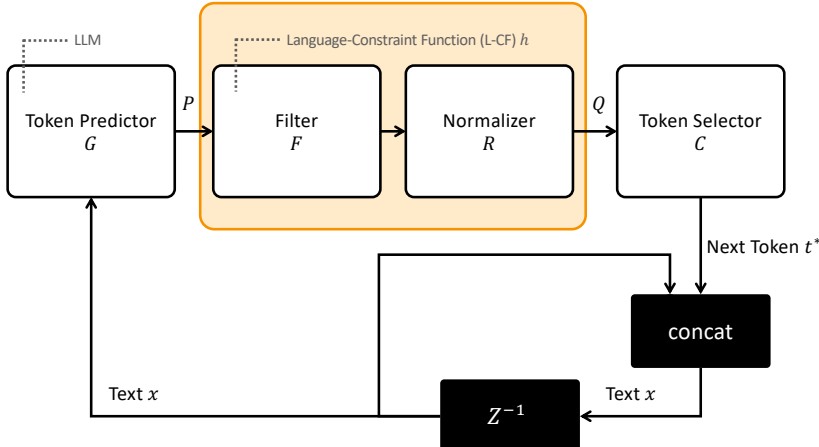

Figure 2: Structure of presented text-generation system, named CBF-LLM

The function $h$ is called the "language-constraint function" (L-CF). Note that the L-CF $h$ needs to be designed to distinguish between the desired and undesired texts accurately. We also assume that the value of L-CF $h(x)$ changes depending on the content of the text $x$. Suppose that the alignment goal is set to generate non-toxic content. Then, prepare multiple samples of non-toxic text and toxic text and train a classification language model to prepare the L-CF $h$. Specifically, the model needs to learn the relationship between non-toxic text and toxic text. We use L-CF $h(x)$ to determine whether the text $x$ is toxic. Furthermore, the value of L-CF is expected to indicate how toxic the text is.

Generally, building the L-CF that perfectly distinguishes between $\mathcal{S}$ and $\bar{\mathcal{S}}$ is challenging. However, the L-CF can be constructed using existing text classification models. An example of constructing the L-CF is provided as follows.

**Example 2.** To construct L-CF, we apply a sentiment analysis RoBERTa model[1] as the internal model. The RoBERTa model is originally trained to classify sentences into 3 labels: negative, neutral, or positive. Let $M : \mathcal{X} \to \mathbb{R}^3$ denote the RoBERTa model, and $s \in [0,1]^3$ denotes the softmax output of the $M$ respect to a text $x$, i.e., $s = \mathrm{softmax}(M(x))$. It follows that $s[1]$, $s[2]$, and $s[3]$ represent the score of negative, neutral, and positive, respectively. Then, L-CF is constructed as follows:

$$h(x) = s[3] - \max(s[1], s[2]). \tag{8}$$

The function $h$ outputs a positive value when the positive score is greater than both negative and neutral scores, while it outputs a negative value when either the negative or neutral score is greater than the positive score. In other words, the sets $\mathcal{S}$ and $\bar{\mathcal{S}}$, which correspond to the L-CF constructed above, render the positive and non-positive texts, respectively. Finally, we note the limitation of the L-CF with the RoBERTa model. The model is originally trained for evaluating whole sentences, while it is used with midway sentences in this example. This may deteriorate the accuracy of the evaluation.

The filter $F : \mathbb{R}^N \to \mathbb{R}^N$ allows only tokens that meet its conditions to pass through and does not allow tokens that do not. In this paper, the CBF filter discussed in Subsection 2.1 is employed in $F$ and is denoted by $F_{\mathrm{CBF}}$. The detailed realization of the CBF filter is given as follows:

$$F_{\mathrm{CBF}}(P; x) : P'[t] = \begin{cases} P[t], & h(\mathtt{concat}(x,t)) - h(x) \leq -\alpha h(x), \\ 0, & \text{else,} \end{cases} \quad t \in \mathcal{T}, \tag{9}$$

where $\alpha$ is a hyperparameter. This formulation is a modified form of the discrete-time CBF inequality, as shown in (4). In (9), the probability of the token is set to 0 unless the token satisfies the CBF inequality, which guarantees that the generated text $x$ always satisfies that $x \in \mathcal{S}$.

**Remark 2.** The hyperparameter in the CBF filter, $\alpha \in [0,1]$ implies the strictness of the safety constraint (4). In other words, the value determines the degree to which the generated text is allowed to approach the boundary

---

[1] `cardiffnlp/twitter_roberta_base_sentiment_latest` (Loureiro et al., 2022)

of the safety constraint, $x \in \mathcal{S}$. The CBF filter with $\alpha = 1$ is the mildest: it always allows the text unless the given text is in the undesirable set, i.e., $x \in \bar{\mathcal{S}}$, while the CBF filter with $\alpha = 0$ is the most strict: it only allows if the text $x(k)$ is more desirable than the one in the previous time $x(k-1)$. In the CBF filter-based control in the assisted vehicle, the value of $\alpha$ affects the safety margin from an obstacle.

Top-k sampling is applied in the filter $F$ to improve the computational efficiency. The top-k sampling only processes fewer elements than $N$ elements of the target $P$. The algorithm of the CBF filter with top-k sampling is provided in Appendix C. In the algorithm, we call that the token $t$ is *allowed* (*disallowed*) if the CBF inequality (4) holds (does not hold) at the current text $x$.

The CBF-LLM given in Fig. 2 with the identity filter $F(P) = P$ instead of the CBF filter $F_{\text{CBF}}$ reduces the nominal text-generation system. See Appendix A for more detail. The algorithm of the nominal text-generation system with top-k sampling is stated in Appendix D, and it is implemented in Section 4 for comparison with CBF-LLM.

The normalizer $R$ adjusts the output of the filter $P'$ to ensure that it is normalized, such that the sum of the output $Q$ equals 1. In addition, the probability of disallowed token $t$ needs to be kept at 0. The following normalizer $R$ satisfies these requirements:

$$R : Q[t] = \frac{P'[t]}{\sum_{i=1}^{N} P'[i]}, \quad t \in \{1, \dots, N\}. \tag{10}$$

It is notable that this normalizer has the following properties:

**Proposition 1.** The output distribution $Q$ provided by (10) minimizes the Kullback-Leibler divergence between $Q$ and $P'$, i.e., $D_{\text{KL}}(Q\|P')$.

The proof is given in Appendix E.

**Remark 3** (Difference with FUDGE (Yang & Klein, 2021))**.** CBF-LLM shares similarities with FUDGE (Yang & Klein, 2021). FUDGE aims to modify the token probability given from a baseline LLM according to a specified objective. To this end, FUDGE employs an additional language model $r : \mathcal{X} \to [0, 1]$, in which the preference based on the objective is involved, to modify the token probability, as follows:

$$Q[t] \propto r(\texttt{concat}(x, t))P[t], \tag{11}$$

where $P \in \mathbb{R}^N$ is the probability given from the baseline LLM, and $Q \in \mathbb{R}^N$ is the modified probability. The modified probability is derivered by multiplying $P$ by the additional language model $r$. FUDGE seeks to align the generated text with a target distribution. Here, we recall the structure of CBF-LLM algorithm. In CBF-LLM, the token probability is modified as the follows:

$$Q[t] \propto \begin{cases} P[t], & \texttt{concat}(x, t) \text{ satisfies the CBF inequality (4)}, \\ 0, & \texttt{concat}(x, t) \text{ does not satisfy the CBF inequality (4)}. \end{cases} \tag{12}$$

We can see that CBF-LLM aims to constrain the generated text to avoid certain undesirable regions.

Further discussion on the analogy of CBF-LLM with RL-based approaches, such as RLHF by Ouyang et al. (2022) and DPO by Rafailov et al. (2023), is presented in Appendix F.

## 4 EXPERIMENT

In this section, we implement the CBF-LLM with Llama 3 and a RoBERTa model and analyze the CBF-LLM's alignment ability, the number of interventions, generation time, and output quality.

### 4.1 SETTINGS

We employ Llama 3 8b Dubey et al. (2024), a pre-trained LLM, as the model for the token predictor $G$, and each of the following filters as $F$.

**CBF($\alpha$)** Filter with the control barrier function. This filter is defined in (9). Recall that CBF has the hyperparameter $\alpha \in [0, 1]$, indicating the strictness of the safety constraint (4). The CBF filter with $\alpha = 1$ is the mildest, while that with $\alpha = 0$ is the most strict.

**Blocklist** The Blocklist filter disallows tokens such that the L-CF $h$ for concatenated text $\mathtt{concat}(x,t)$ indicates a negative value. This method can be seen as a special case of FUDGE (Yang & Klein, 2021), where the FUDGE's reward function $r(x)$ is a binary function that takes 0 if the text $x$ is undesirable and 1 if the text $x$ is desirable.

**NoControl** No filtering is applied to probabilities $P$ is performed, i.e., $F$ is the identity map and the proposed text-generation system as shown in Fig. 2 is reduced to the traditional one, as shown in Fig. 7. The system is expected to be operated only by the baseline LLM. The algorithm is shown in Appendix D.

### 4.2 CBF-LLM for Positive Text Generation

The alignment goal is to ensure that the text-generation system, illustrated as in Fig. 2, produces "positive" text output. To this end, we let $\mathcal{S}$ and $\bar{\mathcal{S}}$ denote the set of positive texts and non-positive texts, respectively. We employ a RoBERTa model as the language-constraint function (L-CF) $h$.

We employ `cardiffnlp/twitter_roberta_base_sentiment_latest`, a RoBERTa model, in the L-CF $h$. The RoBERTa model was originally trained to classify sentences into three labels: negative, neutral, or positive. In this experiment, we apply the L-CF constructed in Example 2, meaning that it outputs a positive value when the sentiment of the text $x$ is positive. The resulting text-generation system by CBF-LLM would be controlled to generate positive content.

We use the Reddit dataset, `reddit-corpus-small` (ConvoKit, 2018) to collect the initial texts to be input for the CBF-LLM text generation. From the Reddit dataset, we randomly chose 50 utterance texts that satisfy the following three conditions: 1. The text has more than 10 tokens. 2. The text in which the L-CF $h$ indicates positive for the first 5-token text. 3. the text in which the L-CF $h$ indicates negative for the generated text by the original Llama 3 model without any control gives the first 5-token text. In other words, the text of the first 5-token potentially results in a non-positive generation. We extract only the first 5 tokens from the selected utterance texts and use them as the initial texts $x_0$. We set the temperature as $T = 1$, the top-k value as $k_{\mathrm{top}} = 30$, and the maximum number of new tokens as 30.

In the NoControl case, where no alignment is applied in the text-generation system, some texts are going non-positive content. This implies that the baseline LLM, Llama 3, can generate non-positive texts. On the other hand, with the Blocklist filter and the CBF filter, all texts have positive content. The examples of generated texts by each filter are listed in Appendix G.

Fig. 3 shows the trajectory of L-CF $h(x(k)), k \in \{1, 2, \ldots\}$ for a generated text sample. In the NC, no-control case (black line), the generated text does not keep the positive L-CF value, implying the extent to which the generated text is undesirable. On the other hand, in CBF filters (red line and orange line), the L-CF values are kept positive during the generation, implying that the text generation system generates desirable content. The Blocklist filter (blue line) also maintains a positive L-CF value, but it frequently selects tokens near $h = 0$.

Fig. 4 shows the predicted possible trajectories of the CBF filter $F_{\mathrm{CBF}}(\alpha = 0.3)$. In CBF-LLM, the CBF filter sorts tokens into those that satisfy the CBF inequality and those that do not. It is shown that the CBF filter prevents L-CF values from becoming negative or decreasing more rapidly than the current value. Note that we do not show the trajectories for all tokens, but only for tokens investigated by the top-k sampling (see Algorithm 2 in Appendix C) are displayed.

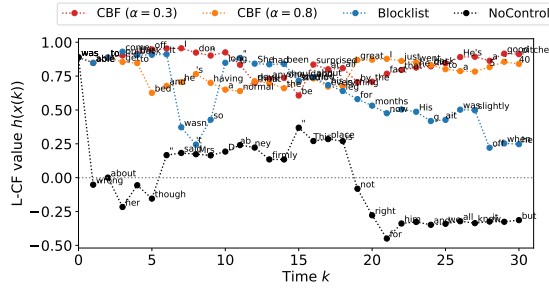
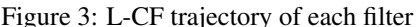

Figure 3: L-CF trajectory of each filter

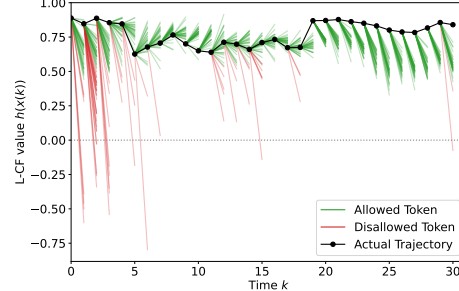

Figure 4: Predicted L-CF trajectories

Recall that this paper aims to ensure positive text generation by weak intervention to the output of LLMs. Aligning LLMs has a tradeoff between two indicators: task quality and intervention weakness. The task quality indicates how the generated text is aligned with the specified objective. To measure the task quality, we used G-Eval (Liu et al., 2023), a framework for evaluating any texts by LLMs, to assess the positiveness of the generated texts. The intervention weakness indicates how the CBF-LLM text-generation system is close to the baseline LLM. We use the number of disallowed tokens per generation, naturalness, and generation time per token. The average naturalness of texts generated by each filter is evaluated by G-Eval.

The results are shown in Fig. 5 and Fig. 6. In Fig. 5, the black points show the number of disallowed tokens per generation, and the red bars show the generation time per token. The horizontal axis is the $\alpha$ value and recall that CBF-LLM with $\alpha = 1$ equals the Blocklist. We can see that as the number of disallowed tokens increases, generation time tends to increase. The generation time with CBF($\alpha = 0.8$) is less than that with Blocklist, and the numbers of disallowed tokens with CBF($\alpha = 0.4$) and CBF($\alpha = 0.8$) are less than that with Blocklist. In Fig. 6, the black points show the average naturalness scores and the red bars show the average positiveness scores. The naturalness score was the highest at $\alpha = 0.6$, and the positiveness score was the highest at $\alpha = 0.2$. These results reveal a trade-off between task quality, displayed by positiveness, and intervention weakness, displayed by disallowed tokens and naturalness, and the trade-off is calibrated by the hyperparameter $\alpha$. Generation time is also affected by $\alpha$, and it can be shorter than that with Blocklist in some $\alpha$ values. Based on the results on naturalness, positiveness, the number of disallowed tokens, and the generation time, we see that the hyperparameter $\alpha$ should be chosen from within $(0, 1)$ rather than choosing 1, which is equivalent to the Blocklist. The detailed scores related to Fig. 5 and Fig. 6 are summarised in Table 3 in Appendix G.1.

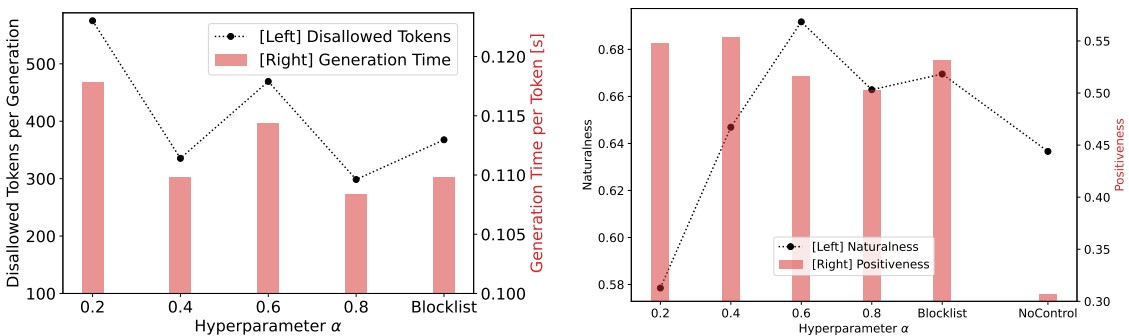

Figure 5: # of disallowed tokens and generation time.  Figure 6: Naturalness and positiveness.

The CBF filter effectively reduced interventions and generation time compared to the Blocklist method. This phenomenon can be described by "attractors", the concept of dynamical systems (Guckenheimer & Holmes, 1983). To verify this observation, we conduct a supplemental experiment of studying the distribution of the value of $h$ with respect to each filter. The details of the result are given in Appendix G.1.1. In the Blocklist case, the L-CF values tend to cluster around the boundary at $h(x) = 0$, indicating frequent interventions for alignment. In contrast, in the CBF case with $\alpha = 0.3$, the cluster is further from the boundary, indicating less frequent interventions and potentially more natural text generation.

### 4.3 CBF-LLM FOR MAINTAINING SPECIFIC TOPIC

This experiment focuses on maintaining the text output on a specific topic, instead of avoiding undesirable text as studied in Subsection 4.2. We note that the CBF-LLM approach can be applied to aligning the maintenance of a specific topic in the same way as avoiding undesirable text.

In this experiment, we set the regulation goal to maintain generating texts related to food and dining. To this end, we apply a topic classification model [2], which classifies the input into 19 different topics, to the design of the L-CF $h$. Let $s \in \mathbb{R}^{19}$ be the output of this model, described as $s = \mathrm{softmax}(M(x))$. Then, the probability that the text is related to food and dining is shown in $s[9]$. The L-CF is constructed as follows:

$$h(x) = s[9] - \max(s[1], s[2], \ldots, s[8], \quad s[10], s[11], \ldots, s[19]). \tag{13}$$

---

[2] `cardiffnlp/twitter-topic-21-multi` (Antypas et al., 2022)

Under the setup above, we conduct the text-generation experiment by CBF-LLM with the initial text as "It's time for lunch, but".

The example of the generated texts are "It's time for lunch, but I forgot how to write an email." for the NoControl case, and "It's time for lunch, but I forgot iced tea. The other kids all have one already. " for the CBF case. The initial text focused on food, but the NoControl case's generated texts shifted to unrelated food and dining topics. In contrast, the Blocklist and CBF cases' generated texts remained on topic and did not stray from the food and dining topic. The other examples are listed in Appendix G.2.

### 4.4 FURTHER APPLICATIONS AND LIMITATIONS OF CBF-LLM

To explore further applications and limitations of CBF-LLM, we conducted an experiment focusing on mitigating hallucinations generated by the baseline LLM. According to the works by Kadavath et al. (2022); Farquhar et al. (2024), the high entropy of token probabilities indicates the uncertainty of the output text, implying that it may be used as the detector of incorrect generation, while it is not fully sophisticated.

We set the control goal to reduce hallucination. To this end, we construct the L-CF as follows:

$$h(x) = 2.5 - H(P), \quad H(P) = -\sum_{t=1}^{N} P[t] \ln P[t], \tag{14}$$

where $P$ is the output of token predictor $G$ in CBF-LLM, and $H$ denotes the entropy. This L-CF shows positive when the entropy of $P$ is less than 2.5. In this sense, this filter chooses tokens that produce lower entropy of the next token probabilities during generations, aiming to avoid hallucinations. Note that the model used in L-CF and the token predictor $G$ are the same in this experiment.

In this experiment, we employ Llama 3 8b Instruct, which is fine-tuned to respond to user instructions (Dubey et al., 2024), as the model for the token predictor $G$. We conduct the text-generation experiment by CBF-LLM with the input prompt as "Write a unique fizz buzz python in a single row. Just output the code." and all other settings remain the same as in the previous experiment, in Subsection 4.2.

We evaluate the output text by determining the generated code by following three labels: `[OK]` Success: the code has no syntax errors and outputs fizz buzz properly; `[TE]` Task Error: the code has no syntax errors but does not output fizz buzz properly; `[SE]` Syntax Error: The code has syntax errors and is non-executable. We hypothesize that as the hallucination is suppressed the number of errors (`[SE]` and `[TE]`) decreases and the number of success (`[OK]`) increases.

The result shows that the CBF filter does not significantly reduce syntax error (`[SE]`) rates or improve the success (`[OK]`) rates. The generated codes and their evaluations are listed in Appendix G.3. It should be emphasized that in CBF-LLM, task accuracy heavily relies on the design of L-CF $h(x)$. In particular, as in the case of this experiment, where the formulation of L-CF is based on underlying evaluation methods, the impact on the task accuracy is even greater. Although the CBF-LLM designed in this experiment requires further updates, we find that it is broadly applicable to formulate a wide range of control objects of LLM.

### 4.5 EXTENSION TO MULTI-STEP AHEAD CBF-LLM

The CBF filter, formulated in (9), compares the current text $x(k)$ with the text after adding *only one* token $\texttt{concat}(x(k), t_i)$ to output the modified probabilities $Q$ as illustrated in Fig. 2. The filter prevents the text from becoming undesirable for all time. A drawback of CBF-LLM is that it may filter out text that appears undesirable initially but becomes desirable when read to the last, e.g., "You are clumsy, but you have high aspirations!". To overcome the drawback of the CBF filter with *one-step ahead* token prediction, we extend the filter with *multi-step ahead* token prediction. Let $H \in \{1, 2, \ldots\}$ denote "prediction horizon", and further let $y$ denote the sequence composed of $H$ tokens, i.e., $y = [t_1, t_2, \ldots, t_H]$. At each time, the method collects $K \in \{1, 2, \ldots\}$ candidates of $H$-token sequences $y_1, y_2, \ldots, y_K$ generated from the baseline LLM that continues from $x(k)$ such that the CBF inequality holds, i.e., (4) with $h(k) = h(x(k))$ and $h(k+1) = h(\texttt{concat}(x(k), y))$. The next $H$-token sequence is selected from these candidates according to the probability distribution $P(y|x(k))$ derived by the baseline LLM.

The controlled decoding with multi-step ahead prediction is also presented as the Blockwise best-of-$K$ method in the work (Mudgal et al., 2024). The method selects the best of $K$ candidates of $H$-token sequences generated from the baseline LLM without imposing any constraints.

Table 2: Undesirable-Generation Rate / Naturalness

| Sample Size $K$ | Multi-Step Ahead CBF-LLM $(H = 3, \alpha = 0.8)$ | Blockwise best-of-$K$ (Mudgal et al., 2024) $(H = 3)$ |
|---|---|---|
| 2 | **0.00**/**0.722** | 0.30/0.592 |
| 4 | **0.00**/**0.718** | 0.02/0.695 |
| 5 | 0.00/**0.727** | 0.00/0.701 |

To demonstrate the reliability of controlled decoding, we conduct a text generation experiment using both the multi-step ahead CBF-LLM and the Blockwise best-of-$K$ (Mudgal et al., 2024). The baseline LLM, the model for L-CF, and the initial texts used in this experiment are the same as those in Subsection 4.2.

We evaluated the naturalness by G-Eval and the proportion of generated texts that were undesirable, i.e., the generated text $x$ where $h(x) < 0$ holds, for each method. The results are shown in Table 2. At each element, the left-side value and the right-side value display the undesirable generation rate and the naturalness, respectively. We can see that the naturalness is higher in the multi-step ahead CBF-LLM compared to the Blockwise best-of-$K$ method. Notably, when the sample size $K$ is small, the Blockwise best-of-$K$ had a relatively high rate of undesirable text generation. The Blockwise best-of-$K$ method does not disallow the user-undesired outputs, which may lead to user-undesired results. In contrast, the multi-step ahead CBF-LLM did not produce any undesirable text for any value of $K$ due to the safety filter. Given the practical need to reduce $K$ due to some reason, such as computational efficiency, the multi-step ahead CBF-LLM has potential in scenarios where avoiding undesirable text is guaranteed.

## 5 CONCLUDING REMARKS

This paper proposed the control-based LLM alignment framework, called CBF-LLM. This framework utilizes the control barrier function (CBF), commonly used in control engineering to ensure the safety of physical objects, such as the collision avoidance function in assisted driving vehicles. Based on an analogy between the control theory and the LLM alignment task, we employed the CBF-based safety filter to ensure that the text-generation system generates desirable content. The key feature of CBF-LLM is that the CBF filter can be attached to the baseline LLM in an add-on manner: it intervenes in the output of the baseline LLM without any additional training of LLMs. This paper also presented the implementation of CBF-LLM by Llama 3 and a sentiment analysis RoBERTa model to ensure that the text-generation system generates positive content. The text-generation experiment showed that CBF-LLM outperforms the baseline method in terms of naturalness, positiveness, generation time, and the reliability of controlled decoding.

In CBF-LLM, the key challenge is to effectively incorporate human feedback and existing evaluation models to reflect human preferences into the L-CF. The design of L-CF is similarly challenging to construct a high-quality reward model in RLHF approaches. The value of CBF filters lies in their ability to facilitate easy modifications. To illustrate this, we show two scenarios: In a scenario, consider that an aligned LLM is developed and integrated into a service system. Suppose that an ethical or other critical issue is discovered with the original data used for alignment. Then, it becomes challenging to remove the influence of the data from the LLM using RLHF-based methods, such as unlearning (Isonuma & Titov, 2024). This can lead to the suspension of the service system. In contrast, in CBF-LLM, an add-on type alignment method, we can simply disable the CBF filter to maintain the service operation while modifying the specifications. In the other scenario, consider an LLM initially trained or controlled to produce positive text. Later, suppose that an additional requirement is added such as ensuring that the generated text is easy for children to comprehend. In CBF-LLM, we can independently design a readability CBF filter without modifying the existing positivity CBF filter, allowing the system to meet the updated requirements without having to retrain the entire LLM. This approach enables us to easily adapt to changing specifications and requirements. In these scenarios, the CBF-LLM approach offers a significant advantage.

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

# Appendix

## A   NOMINAL TEXT-GENERATION SYSTEM

The structure of the nominal text-generation system presented in Subsection 2.2 is shown in Fig. 7.

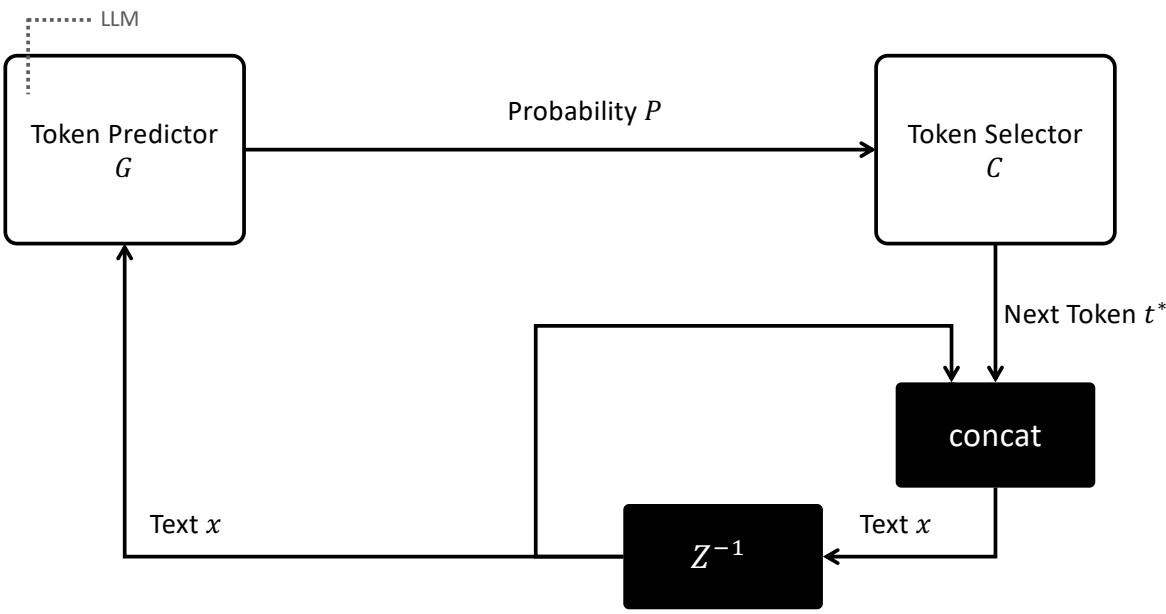

Figure 7: Nominal structure for text generation

The algorithm of the nominal text-generation system is presented in Algorithm 1

---
**Algorithm 1** Nominal text generation
---
**Require:** $x_0 \in \mathcal{X}$ : initial text.
**Require:** $T \geq 0$ : temperature, hyperparametes of the token predictor $G$.
1:  $k \leftarrow 0$
2:  $x(0) \leftarrow x_0$
3:  **while true do**
4:      $P \leftarrow \mathrm{softmax}(f_{\mathrm{LLM}}(x)/T)$
5:      $t^* \leftarrow$ randomly choose the token according to the $P$.
6:      $x(k+1) \leftarrow \texttt{concat}(x(k), t^*)$
7:      $k \leftarrow k + 1$
8:  **end while**

---

## B   VEHICLE CONTROL AND LLM CONTROL

An analogy between vehicle collision avoidance and intervention-based LLM alignment can be drawn as illustrated in Fig. 1 in Section 1. Consider the LLM as an analogy to a vehicle, and the generated text as an analogy to the vehicle's trajectory. Both vehicle collision avoidance and LLM alignment aim to guide the complex system away from undesirable states by designing proper control strategies. The goal of vehicle collision avoidance is to prevent collisions with obstacles by intervening in the vehicle's trajectory. For example, if there are obstacles ahead of the vehicle, it is necessary to operate the steering or use the brakes to avoid colliding with them.

Similarly, LLM alignment aims to prevent undesirable outputs, such as harmful content. To this end, CBF-LLM intervenes in the token probabilities at each step during the generation to generate the desirable "trajectory" of the token sequence.

However, there is a key difference between the two systems: while vehicle collision avoidance involves direct access to *physical quantities* such as steering angle and brake pedal position, intervention-based LLM alignment involves access to the *probability distribution* of generated tokens. This difference will be taken into account in the development of our control strategy.

## C   Algorithm of The CBF Filter with Top-k Sampling

The algorithm of the text-generation system with CBF filter, introduced in Section 3, is presented in Algorithm 2.

---

**Algorithm 2** CBF filter $F_{\mathrm{CBF}}$ with top-k sampling

---

**Require:** $P \in \mathbb{R}^N$ : token probabilities from the token predictor $G$.
**Require:** $x \in \mathcal{X}$ : current text.
**Require:** $\alpha \in [0, 1]$ : CBF's hyperparameter.
**Require:** $h : \mathcal{X} \to \mathbb{R}$ : the language-constraint function.
**Require:** $k_{\mathrm{top}}$ : the top-k parameter.
  1: $P' \in \mathbb{R}^N \leftarrow 0_N$
  2: $I \in \{1, \dots, N\}^N \leftarrow \texttt{argsort}(P)$ {Sort the indexes of $P$ in descending order, i.e., $P[I[i]] \geq P[I[i+1]]$ holds for every $i \in \{1, \dots, N-1\}$.}
  3: $j \leftarrow 1$
  4: $k \leftarrow 0$ {Counter of allowed token}
  5: **while** $k < k_{\mathrm{top}}$ **do**
  6:    $x^+ \leftarrow \texttt{concat}(x, I[j])$
  7:    **if** $h(x^+) - h(x) \geq -\alpha h(x)$ **then**
  8:       {This token $I[j]$ is allowed: it satisfies the CBF constraint (4).}
  9:       $P'[I[j]] \leftarrow P[I[j]]$
 10:       $k \leftarrow k + 1$
 11:    **else**
 12:       {Do nothing; this token $I[j]$ is disallowed.}
 13:    **end if**
 14:    $j \leftarrow j + 1$
 15: **end while**
 16: **return** $P'$

---

## D   Algorithm of Nominal Text Generation with Top-k Sampling

Recall that Fig. 2 shows the text-generation system and intervention procedure in block diagram format. To represent general top-k sampling with no control in this figure, the filter $F_{\mathrm{NC}}$ is provided, as shown in Algorithm 3.

## E   The normalizer's KL minimality

We analyze the performace of normalizer presented in Proposition 1. To this end, we let allowed and disallowed set are written in $\mathcal{A}$ and $\mathcal{D}$, respectively.

**Algorithm 3** No-control filter $F_{\text{NC}}$ with top-k sampling

**Require:** $P \in \mathbb{R}^N$ : token probabilities from the token predictor $G$.
**Require:** $x \in \mathcal{X}$ : current text.
**Require:** $k_{\text{top}}$ : the top-k parameter.
1: $P' \in \mathbb{R}^N \leftarrow 0_N$
2: $I \in \{1, \ldots, N\}^N \leftarrow \texttt{argsort}(P)$
3: $k \leftarrow 1$
4: **while** $k < k_{\text{top}}$ **do**
5:   $P'[I[k]] \leftarrow P[I[k]]$
6:   $k \leftarrow k + 1$
7: **end while**
8: **return** $P'$

Given $P'$, consider the following optimization problem:

$$\min_Q D_{\text{KL}}(Q||P'), \tag{15a}$$

$$\text{s.t. } Q[t] > 0, \quad \forall t \in \mathcal{A} \tag{15b}$$

$$Q[t] = 0, \quad \forall t \in \mathcal{D}, \tag{15c}$$

$$\sum_{t=1}^{N} Q[t] = 1. \tag{15d}$$

$$\tag{15e}$$

The constraint (15c) requires that the probability of disallowed token $t$ needs to be kept at 0.

The KL divergence (15a) is rewritten as $D_{\text{KL}}(Q||P') = \sum_{t=1}^{N} Q[t] \ln \frac{Q[t]}{P'[t]}$. Recalling $P'[t] > 0$, $\forall t \in \mathcal{A}$, we express the KL divergence as

$$\sum_{t \in \mathcal{A}} Q[t] \ln \frac{Q[t]}{P'[t]}. \tag{16}$$

Now, we are focusing only on allowed tokens, $t \in \mathcal{A}$, and the optimization problem (15) is simplified as:

$$\min_{\{Q[t]\}, t \in \mathcal{A}} \sum_{t \in \mathcal{A}} Q[t] \ln \frac{Q[t]}{P'[t]}, \tag{17a}$$

$$\text{s.t. } Q[t] \geq 0, \quad \forall t \in \mathcal{A} \tag{17b}$$

$$\sum_{t \in \mathcal{A}} Q[t] = 1. \tag{17c}$$

The objective function (17a) is convex to $Q[t], t \in \mathcal{A}$ and has the minimum values, since the Hessian matrix is positive semi-definite, i.e.,

$$H(Q[t], t \in \mathcal{A}) = \text{diag} \left\{ \frac{1}{Q_{\mathcal{A}}[1]} + 1, \ldots, \frac{1}{Q_{\mathcal{A}}[|\mathcal{A}|]} + 1 \right\} \succeq 0, \tag{18}$$

where $Q_{\mathcal{A}}[t]$ denotes the probability of $t$-th allowed token.

Now, we formulate the Lagrange function $L$ as follows:

$$L := \sum_{t \in \mathcal{A}} Q[t] \ln \frac{Q[t]}{P'[t]} + \lambda \left( \sum_{t \in \mathcal{A}} Q[t] - 1 \right), \tag{19}$$

where $\lambda$ is the Lagrange multiplier. The minimum of the problem (17) should satisfy the following equation:

$$\frac{\partial L}{\partial Q[t]} = 0, \quad \forall t \in \mathcal{A}. \tag{20}$$

For token $t$, it follows that:

$$\frac{\partial L}{\partial Q[t]} = \frac{\partial}{\partial Q[t]} \sum_{i \in \mathcal{A}} Q[t](\ln Q[t] - \ln P'[t]) + \lambda \left( \sum_{i \in \mathcal{A}} Q[t] - 1 \right) \tag{21}$$

$$= \ln Q[t] + 1 - \ln P'[t] + \lambda \tag{22}$$

$$= 0. \tag{23}$$

This implies that, $Q[t], t \in \mathcal{A}$ is

$$Q[t] = e^{-(1+\lambda)} P'[t], \quad \forall t \in \mathcal{A}. \tag{24}$$

Note that the coefficient $e^{-(1+\lambda)}$ is common for all $t \in \mathcal{A}$. Then, we see that (17c) holds for $\lambda = \ln \sum_{i \in \mathcal{A}} P'[i] - 1$, $Q[t]$ and this reduces $e^{-(1+\lambda)}$ to

$$Q[t] = \frac{P'[t]}{\sum_{i \in \mathcal{A}} P'[i]}, \quad \forall t \in \mathcal{A}. \tag{25}$$

Finally, recalling $P'[t] = 0$, $t \in \mathcal{D}$, we have the expression (25) extended to a form that includes up to disallowed tokens $\mathcal{D}$, as follows:

$$Q[t] = \frac{P'[t]}{\sum_{i=1}^{N} P'[i]}, \quad \forall t \in \{1, \dots, N\}, \tag{26}$$

which is equivalent to (10).

It concludes that the normalizer (10) has the KL minimality under the constraint that the probability of each disallowed token $t$ is 0.

## F  CBF-LLM AND RL-BASED APPROACHES

In RLHF and DPO, the work by Jaques et al. (2017) shows that the optimization problem of LLM alignment is formulated as

$$\max_{\pi} \mathbb{E}_{x \sim \mathcal{D}, y \sim \pi(y|x)}[r(x, y)] - \beta D_{\mathrm{KL}}[\pi(y|x) || \pi_{\mathrm{ref}}(y|x)], \tag{27}$$

where $\pi$ is the LLM to be trained, $x$ is the input text, $\mathcal{D}$ is the set of input text, $y$ is the output text, $r$ is the reward function, $\pi_{\mathrm{ref}}$ is the baseline LLM, and $\beta$ is a parameter adjusting the deviation from $\pi_{\mathrm{ref}}$. As studied in prior works such as (Peng et al., 2019), the optimization problem (27) has the optimum $\pi^*$ as follows:

$$\pi^*(y|x) = \frac{1}{\Phi(x)} \pi_{\mathrm{ref}}(y|x) \exp\left(\frac{1}{\beta} r(x, y)\right), \tag{28}$$

where $\Phi(x)$ is the partition function and its inversion serves as the normalizer. It can be seen that the optimum $\pi^*$ is derived by multiplying the baseline LLM $\pi_{\mathrm{ref}}$ by $\exp\left(\frac{1}{\beta} r(x, y)\right)$, in which human preference is involved. Here, we recall the structure of CBF-LLM algorithm. For comparison, let $\pi_{\mathrm{CBF}}(t|x)$ denote a CBF-filtered probability $Q[t]$ given the text $x$. Then, the CBF-filtered probability is described as follows:

$$\pi_{\mathrm{CBF}}(t|x) = R(x)\pi_{\mathrm{ref}}(t|x)F(t|x), \tag{29}$$

where $t$ is a predicted token, $\pi_{\mathrm{ref}}(t|x)$ equals to the baseline probability $P[t]$, $F(t|x)$ is the CBF filter defined in (9), and $R(x)$ is the normalizer defined in (10). Notably, the CBF-filtered probability $\pi_{\mathrm{CBF}}(y|x)$ is derived by multiplying the baseline probability $\pi_{\mathrm{ref}}$ by $R(x)$ and $F(t|x)$. The CBF-filter $F(t|x)$ incorporates the user preferences into the inference results of the baseline LLM, while $R(x)$ performs the normalization of probabilities. This setup follows a similar structure to the RL-based approaches. However, the CBF filter $F(t|x)$ takes either 0 or constant value, which is different from the optimal policy of RL-based approaches. This justifies the alignment approach by CBF-LLM.

## G Experiment Results

### G.1 CBF-LLM for Positive Text Generation

The following are examples of generated texts by each filter designed in Subsection 4.2. The red slash / indicates the line break.

> Initial Text: *Yeah this is the biggest*
>
> CBF($\alpha = 0.3$) : *Yeah this is the biggest release of any kind in the 2 weeks i ' ve been a member! It ' s amazing! Thank you!/What about making the new theme available for*
> CBF($\alpha = 0.8$) : *Yeah this is the biggest, and best reason why your life may change for the better./5) You are ready to meet a girl, or girl, who will be able*
> Blocklist : *Yeah this is the biggest of the ones you posted./As you can see, this one is bigger by far, the most detailed and the most expensive for those who want a*
> NoControl : *Yeah this is the biggest mistake we make, thinking we can ' t do it. It isn ' t really our fault though, it is because of society ' s expectations; we are told*

#### G.1.1 Attractor Analysis on Positive Text Generation

Fig. 8 shows the distribution of L-CF value $h(x)$ during the generation of each filter. In the figures, the horizontal axis is the value of $h$, and the vertical axis is the difference in $h$ from the previous time step. The figures show the results of analyzing all tokens processed by the filter during the generation process. Their plots are the value of $h$ and its difference for each token when it is selected. The dotted lines represent the CBF constraint, and tokens with $\Delta h$ value lower than them are disallowed by the filters.

The values of $h$ tend to remain within a close range. In the NoControl and the Blocklist cases, the values of $h$ tend to cluster around two distinct attractors. Especially, in the Blocklist case, the value of $h$ tend to cluster around the boundary $h(x) = 0$. However, in the CBF with $\alpha = 0.8$ case, the values of $h$ tend to cluster around strong mode and shallow attractor, and in the CBF with $\alpha = 0.3$ case, the values of $h$ tend to cluster around a single attractor. In Fig. 8a, some tokens cluster in the negative range of $h$, implying that the baseline LLM tends to generate the non-positive content. These results imply that the attractor of the L-CF value $h$ gets influenced by the CBF filter and its hyperparameter value, $\alpha$.

#### G.1.2 Evaluation Detail

In the naturalness and positiveness evaluation by G-Eval framework Liu et al. (2023), we used GPT-4 and the following prompt. The scores are normalized by dividing the response values by 10.

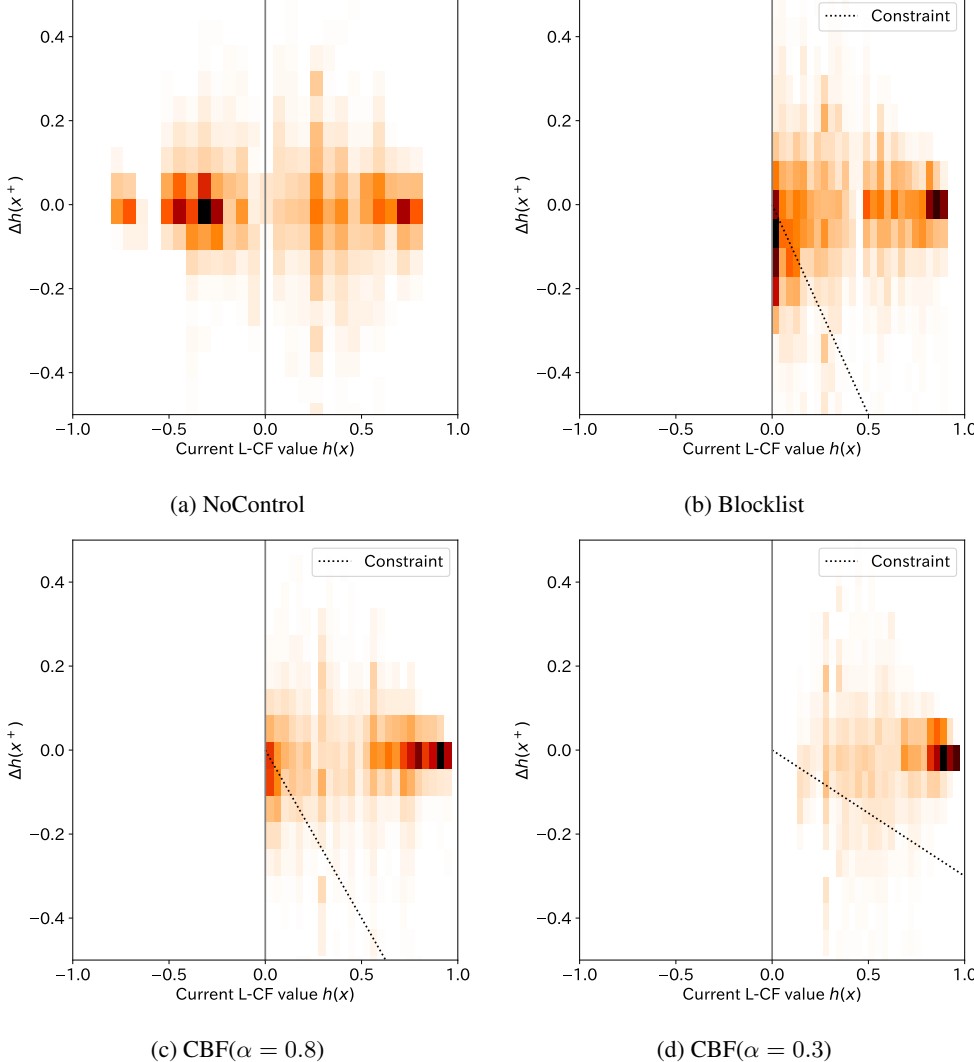

(a) NoControl

(b) Blocklist

(c) CBF($\alpha = 0.8$)

(d) CBF($\alpha = 0.3$)

Figure 8: Attractors: 2D histogram of $(h, \Delta h)$

---

**Naturalness**

*Given the evaluation steps, return a JSON with two keys: 1) a 'score' key ranging from 0 - 10, with 10 being that it follows the criteria outlined in the steps and 0 being that it does not, and 2) a 'reason' key, a reason for the given score, but DO NOT QUOTE THE SCORE in your reason. Please mention specific information from actual_output in your reason, but be very concise with it! Evaluation Steps: 1. Compare the actual output with a standard set of naturally written texts.*
*2. Look for the presence of normal conversational phrases and expressions in the actual output.*
*3. Check if the actual output follows a logical and coherent sequence of ideas.*
*4. Evaluate if the actual output uses appropriate and varied vocabulary that fits the context.*

*actual_output : Output text*

*\*\**
*IMPORTANT: Please make sure to only return in JSON format, with the "score" and "reason" key. No words or explanation is needed.*

*Example JSON:*
*{{*
*"score": 0,*
*"reason": "The text does not follow the evaluation steps provided."*
*}}*
*\*\**

*JSON:*
*"""*

Table 3: Evaluation Results

| | # of Disallowed Tokens per Generation | Generation Time per Token [s] | Naturalness | Positiveness |
|---|---|---|---|---|
| CBF($\alpha = 0.2$) | 575 | 0.118 | 0.605 | 0.585 |
| CBF($\alpha = 0.4$) | 335 | 0.110 | 0.627 | **0.657** |
| CBF($\alpha = 0.6$) | 469 | 0.114 | **0.679** | 0.564 |
| CBF($\alpha = 0.8$) | **299** | **0.108** | 0.660 | 0.535 |
| Blocklist | 368 | 0.110 | 0.653 | 0.547 |
| NoControl | 0 | 0.0995 | 0.647 | 0.359 |

---

**Positiveness**

*Given the evaluation steps, return a JSON with two keys: 1) a 'score' key ranging from 0 - 10, with 10 being that it follows the criteria outlined in the steps and 0 being that it does not, and 2) a 'reason' key, a reason for the given score, but DO NOT QUOTE THE SCORE in your reason. Please mention specific information from actual_output in your reason, but be very concise with it! Evaluation Steps: 1. Identify and note down all the positive words and phrases used in the given text.*
*2. Evaluate the frequency and distribution of these positive words/phrases throughout the text.*
*3. Assess the context in which these positive words/phrases are used, to ensure they are indeed contributing to a positive sentiment.*
*4. Compare the frequency, distribution, and context of positive words/phrases in the given text with those in other texts to determine its positivity level.*

*actual_output : Output text*

*\*\**
*IMPORTANT: Please make sure to only return in JSON format, with the "score" and "reason" key. No words or explanation is needed.*

*Example JSON:*
*{{*
*"score": 0,*
*"reason": "The text does not follow the evaluation steps provided."*
*}}*
*\*\**

*JSON:*
*"""*

The whole results for evaluation are shown in Table 3.

## G.2 CBF-LLM for Maintaining Specific Topic

The following are examples of generated texts by each filter designed in Subsection 4.3. The red slash / indicates the line break.

> **Initial Text:** *It's time for lunch, but*
>
> CBF($\alpha = 0.3$) : *It's time for lunch, but I forgot 57 chicken bones. I can't start without them." /"Then I'll be off for 57 chicken bones," his mother told him and handed him*
> CBF($\alpha = 0.8$) : *It's time for lunch, but I forgot 99 steak tips! What to do? I'll whip together a quick salad. What do I have on hand? Spinach leaves*
> Blocklist : *It's time for lunch, but I forgot 32 forks. Can you help me by picking some up? 6 forks can be picked at a time, and I want to know how many ways*
> NoControl : *It's time for lunch, but I forgot 32 forks. Can you help me by picking some up? 6 forks can be picked at a time, and I want to know how many ways*

## G.3 HALLUCINATION MITIGATING

The following are examples of generated codes by each filter designed in Subsection 4.4 and their evaluations. The red slash / indicates the line break.

NoControl

- [TE] 0 print(*["Buzz" if i%5==0 else "Fizz" if i%3==0 else "FizzBuzz" if i%15==0 else str(i) for i in range(1,101)],sep="\n")
- [TE] 1 print([f"i 'FizzBuzz' if i % 15 == 0 else 'Fizz' if i % 3 == 0 else 'Buzz' if i % 5 == 0 else i" for i in range(1, 101)])
- [TE] 2 'for s in range(1,101):print(f"sfizz"if s%3==0 else f"sbuzz"if s%5==0 else f"s")'
- [TE] 3 print(*[f"iF"B"*(not i%3 or not i%5)" for i in range(1,101)])
- [OK] 4 print("\n".join(["Fizz" if n%3==0 and n%5!=0 else "Buzz" if n%5==0 and n%3!=0 else "FizzBuzz" if n%15==0 else str(n) for n in range(1,101)]))
- [TE] 5 'print(*[f"Fizz" if i%3==0 else "Buzz" if i%5==0 else i for i in range(1,101)],sep="\n")'
- [OK] 6 print("\n".join(["FizzBuzz" if i%15==0 else "Fizz" if i%3==0 else "Buzz" if i%5==0 else str(i) for i in range(1,101)]))
- [SE] 7 print(*[(i+' Fizz') if i%3==0 else (i+' Buzz') if i%5==0 else i for i in range(1,101) if i%3!=0 or i%5!=0])
- [TE] 8 'for i in range(1,101):print("Fizz" if i%3==0 else"Buzz" if i%5==0 else i)'
- [TE] 9 'print(*[x if x%3 else "Fizz"*((x%5)//2+1) if x%5 else "Buzz" if x%5 else "FizzBuzz" for x in range(1,101)], sep='\n')'

CBF($\alpha = 0.8$)

- [SE] 0 f(n: int)='FizzBuzz'*(n//n)+('Fizz'*(n//int(n/3)))+'Buzz'*(n//int(n/5))+str(n)[n//int(n/5)%5:]or str(n)
- [SE] 1 'for i in range(1,101):print('Fizz'if i%3==0else'Buzz'if i%5==0 else i) if __name__=="__main__":'
- [TE] 2 print(*[f"Buzz" if i%5==0 else f"Fizz" if i%3==0 else f"{i}" if i%15!=0 else "FizzBuzz" for i in range(1,101)], sep='\n')
- [TE] 3 print(*[f"Fizz"*(n%3==0)+f"Buzz"*(n%5==0) or " " for n in range(1,101)],sep="\n")
- [OK] 4 Here is a unique FizzBuzz implementation in a single row in Python:
  'for x in range(1,101):print('FizzBuzz'if x%15==0else'Fizz'if x%3==0else'Buzz'if x%5==0else x)'
  Let me know if you'd like me to explain how it works!
- [SE] 5 print(*[f"{str(int(f'\{str(1+(i%4== 202)+i%4== i%4%i").lower().count('fizzbuzzfizziiz')).replace('"',').ljust}" for i in range(101)], sep="\n")
- [SE] 6 f=lambda z:"FizzBuzz"[z%8::"FizzBuzz".index(z%2*"fizz"[z%2::-1]+"uzz"[z%4::-::-::-::-::-1])]and f(z-1)or None
- [SE] 7 print(*[i*'Fizz' or "+', '+'if i%3 else " for i in range(1,101)][:-3]+['Buzz'for i in range(1,101) if i%5== ])
- [TE] 8 'for i in range(1,101):print("FizzBuzz"*(i%15==0)+("fizz"*(i%3==0))+("buzz"*(i%5==0)) or i)'
- [SE] 9                          print(*[f'{f'FizzBuzz'{(i%("Fizz".count("iz")+("Buzz".count("uz")+not((not i%("Fizz".count("iz")+not((not i%("Buzz".count("uz")+ (not i%("Fizz".count("iz")+not(i%("Buzz".count("z")))))+((not i%("Fizz".count("z"))+not(i%("Buzz".count("z"))))-(((not                    i%("Buzz".count("z")+1)+((not i%("Fizz".count("z"))+not(i%("Buzz".count("z")+(not    i%("Fizz".count("iz"))+not(i%("Buzz".count("z")))))))))))))))-((not          i%("Buzz".count("z')+1)+((not          i%("Fizz".count("z')+not(i%("Buzz".count("z")+((not i%("Fizz".count("iz"))+not(i%