# OpenReview forum: "CBF-LLM: Safe Control for LLM Alignment"
_ICLR.cc/2025/Conference — Submitted to ICLR 2025_

### Official Review · Reviewer_2hj6 · 2024-11-01

**Soundness:** 1
**Presentation:** 3
**Contribution:** 2
**Rating:** 3
**Confidence:** 4

**Summary:**

The paper presents a control-based framework for aligning LLMs to ensure the generation of user-desirable text. The framework leverages control barrier functions (CBFs), a concept from control engineering, to intervene in the output generation of LLMs, aiming to prevent the production of harmful, biased, or toxic content. The key contributions of the paper are:

1 This paper presents a novel framework that applies a CBF-based safety filter to LLM output, aiming to reduce the need for interventions in text generation while maintaining alignment with user specifications.

2 This paper demonstrates the practical application of the CBF-LLM framework using Llama 3 and a sentiment analysis RoBERTa model, showing its effectiveness in generating positive content.

3 This paper attempts to connect control engineering with NLP by adapting control theory techniques for LLM alignment, offering a new perspective on ensuring the safety and ethicality of LLM-generated content.

**Strengths:**

1. The paper is well-written, and I really like their figures.
2. There is a detailed theoretical transfer and explanation on control engineering and how it can be extended to LLMs.
3. This paper provides some inspiration for future non-parametric optimization methods for LLMs.
4. To some extent, it achieves an alignment from weak (RoBERTa) to strong (LLaMA).

**Weaknesses:**

1. The experiments and datasets lack persuasiveness. All experiments in the paper are based on only a few queries, with almost no evaluations on common LLM benchmarks or other metric.
2. The approach heavily relies on a Language-Constraint Function (L-CF), implemented with RoBERTa and assumed to be a golden classifier. However, I believe that assuming RoBERTa as an ideal classifier is not reasonable in practical use.

**Questions:**

1. Why not conduct experiments on at least a small-scale widely-used LLM evaluation datasets, such as a subset of Anthropic/hh-rlhf?
2. I feel that finding a reliable Language-Constraint Function (L-CF) is similarly challenging to training a golden reward model. It might be insightful to explore scenarios where the Language-Constraint Function (L-CF) has varying levels of reliability.

---

> ### Author Response · Authors · 2024-11-23
> **Response to Reviewer 2hj6 (1/2)**
>
> We appreciate Reviewer **2hj6** for raising many comments and introducing related datasets.  We have conducted additional text generation experiments using a dataset and provided comparisons with existing works.  The detailed discussion of each comment raised by the reviewer is presented as follows.
>
> ---
>
> # Weakness 1
> We appreciate for the instructive comments.
>
> We conducted additional generation experiments using the Reddit corpus, which was also utilized for the experiments in the paper "Controlled Decoding from Language Models"**[1]**.  We provided 50 samples each to five CBF filters with different hyperparameters, Blocklist, and NoControl, and performed text generation on 300 samples in total.  We evaluated the generated texts using three methods: evaluating positiveness and naturalness using G-Eval **[2]**, measuring the number of disallowed tokens, and measuring generation time per token. The results are summarized in the following table (the same as `Table 3` at `Appendix F.1` in the revised paper):
>
> Method | \# of Disallowed Tokens per Generation | Generation Time per Token [s] | Naturalness | Positiveness
> -|-|-|-|-
> CBF($\alpha=0.2$) | 575 | 0.118 | 0.605 | 0.585
> CBF($\alpha=0.4$) | 335 | 0.110 | 0.627 | $\bf\underline{0.657}$
> CBF($\alpha=0.6$) | 469 | 0.114 | $\bf\underline{0.679}$ | 0.564
> CBF($\alpha=0.8$) | $\bf\underline{299}$ | $\bf\underline{0.108}$ | 0.660 | 0.535
> Blocklist | 368 | 0.110 | 0.653 | 0.547
> NoControl | 0 | 0.0995 | 0.647 | 0.359
>
> Positiveness was highest when using CBF ($\alpha=0.4$), Naturalness was highest when using CBF ($\alpha=0.6$), the number of disallowed tokens was lowest when using CBF ($\alpha=0.8$), and generation time per token was lowest when using CBF ($\alpha=0.8$). Our CBF filters outperformed the Blocklist method, which is a special case of FUDGE **[3]** with a binary-valued reward function, in all metrics.
>
> ---
>
> # Weakness 2
> We agree with this concern. The performance of our approach heavily depends on the quality of the Language-Constraint Function (L-CF). We acknowledge that assuming RoBERTa as a golden classifier may not be reasonable in practical scenarios. To remark the limitation of L-CF constructed with RoBERTa model, we have added the following remark in `CBF-LLM` section in the revised paper, as follows:
> > Finally, we note the limitation of the L-CF with the RoBERTa model.
> The model is originally trained for evaluating whole sentences, while it is used with midway sentences in this example.
> This may deteriorate the accuracy of the evaluation.
>
> For our current approach, our primary effort was dedicated to establishing a foundational framework.  Our future work includes finding and creating language models suitable for L-CF.
>
> ---
>
> # Question 1
> In this paper, we used a pre-trained foundation model (Llama 3 8b) to focus on evaluating the control performance of the baseline decoder. Therefore, we did not use $\texttt{Anthropic/hh-rlhf}$, which is a benchmark for instruct-following models, as they are not directly applicable to this experiment.  Instead, we used the Reddit corpus for the additional experiment. The results are shown in the response to the **Weakness 1**.  We appreciate for introducing datasets for benchmarks.

---

> ### Author Response · Authors · 2024-11-23
> **Response to Reviewer 2hj6 (2/2)**
>
> # Question 2
> We agree with the comment. Creating L-CF for CBF-LLM is as difficult as creating a reward model for RLHF.  We appreciate for the comment of exploring the scenarios.  We discussed some scenarios where using an L-CF to create a CBF-LLM is more beneficial in the `Concluding Remarks` section, as follows:
> > In CBF-LLM, the key challenge is to effectively incorporate human feedback and existing evaluation models to reflect human preferences into the L-CF.
> The design of L-CF is similarly challenging to construct a high-quality reward model in RLHF approaches.
> The value of CBF filters lies in their ability to facilitate easy modifications.
> To illustrate this, we show two scenarios:
> In a scenario, consider that an aligned LLM is developed and integrated into a service system.
> Suppose that an ethical or other critical issue is discovered with the original data used for alignment.
> Then, it becomes challenging to remove the influence of the data from the LLM using RLHF-based methods, such as unlearning.
> This can lead to the suspension of the service system.
> In contrast, in CBF-LLM, an add-on type alignment method, we can simply disable the CBF filter to maintain the service operation while modifying the specifications.
> In the other scenario, consider an LLM initially trained or controlled to produce positive text.
> Later, suppose that an additional requirement is added such as ensuring that the generated text is easy for children to comprehend.
> In CBF-LLM, we can independently design a readability CBF filter without modifying the existing positivity CBF filter, allowing the system to meet the updated requirements without having to retrain the entire LLM.
> This approach enables us to easily adapt to changing specifications and requirements.
> In these scenarios, the CBF-LLM approach offers a significant advantage.
>
> ---
>
> # Reference List
>
> [1] [Controlled Decoding from Language Models](https://icml.cc/virtual/2024/poster/33639), ICML 2024.
>
> [2] [G-Eval: NLG Evaluation using GPT-4 with Better Human Alignment](https://aclanthology.org/2023.emnlp-main.153/), Proceedings of the 2023 Conference on Empirical Methods in Natural Language Processing.
>
> [3] [FUDGE: Controlled Text Generation With Future Discriminators](https://aclanthology.org/2021.naacl-main.276/), Proceedings of the 2021 Conference of the North American Chapter of the Association for Computational Linguistics: Human Language Technologies.

---

### Official Review · Reviewer_qH9r · 2024-11-05

**Soundness:** 3
**Presentation:** 4
**Contribution:** 3
**Rating:** 8
**Confidence:** 4

**Summary:**

The paper presents a method to filter the language produced by an LLM such as LLAMA; the filtering happens via an auxiliary algorithm/model that scans the original LLM’s output probabilities of vocabulary words to be produced next and picks out the word(s) that satisfy an additional requirement such as producing positive or negative text, or text on a specific topic. This filter function/model seems to be another pre-trained LLM as well, e.g., Roberta.

**Strengths:**

The work retrofits LLAMA-type LLM to produce desired constrained language in a general way.

The work has a theoretical component to it wherein it attempts to control  language production using established techniques from the perspective of control theory.

The approach has been used to produce text that suits the purpose under different constraints.

**Weaknesses:**

The presented theoretical and algorithmic control approach seems cumbersome and done in a roundabout way, not straightforwardly.

There seems to be no evaluation of the generated text after filtration. Output sentence examples are given and seem good, but the main paper doesn’t contain evaluation metrics and scores.

I find the analogy with a car not particularly persuasive. In a car, to avoid obstacles, the car’s internal processes must change to produce new trajectory for the car. However, in an LLM, to produce the desired output, the proposed approach lets the LLM produce output as usual, but seems to filter later to suit the purpose.

It will be good to cite a relevant paper: Zingale and Kalita. 2024. Language Model Sentence Completion with a Parser-Driven Rhetorical Control Method. In Proceedings of the 18th Conference of the European Chapter of the Association for Computational Linguistics, pages 193–203.

**Questions:**

NA

---

> ### Author Response · Authors · 2024-11-23
> **Response to Reviewer qH9r**
>
> We appreciate Reviewer qH9r for raising many comments and introducing related works.  We have conducted additional text generation experiments using a dataset and provided comparisons with existing works.  The detailed discussion of each comment raised by the reviewer is presented as follows.
>
> ---
>
> # Weakness 1
> > **Begins with** "The presented their...."
>
> We agree that the theoretical and algorithmic control approach, particularly the explanation of LLM, may seem cumbersome. However, our focus was on elucidating the value of CBF-LLM, and we intentionally devoted our efforts to highlighting its benefits, even if the explanation took a more circuitous route.
>
> ---
>
> # Weakness 2
> > **Begins with** "There seems to be n..."
>
> In the additional experiment, we evaluated the generated texts using three methods: evaluating positiveness and naturalness using G-Eval **[2]**, measuring the number of disallowed tokens, and measuring generation time per token. The  experiments were conducted using the Reddit corpus.  We provided 50 samples each to five CBF filters with different hyperparameters, Blocklist, and NoControl, and performed text generation on 300 samples in total.  The results are summarized in the following table (the same as `Table 3` in `Appendix F.1` of the revised paper)
>
> Method | \# of Disallowed Tokens per Generation | Generation Time per Token [s] | Naturalness | Positiveness
> -|-|-|-|-
> CBF($\alpha=0.2$) | 575 | 0.118 | 0.605 | 0.585
> CBF($\alpha=0.4$) | 335 | 0.110 | 0.627 | $\bf\underline{0.657}$
> CBF($\alpha=0.6$) | 469 | 0.114 | $\bf\underline{0.679}$ | 0.564
> CBF($\alpha=0.8$) | $\bf\underline{299}$ | $\bf\underline{0.108}$ | 0.660 | 0.535
> Blocklist | 368 | 0.110 | 0.653 | 0.547
> NoControl | 0 | 0.0995 | 0.647 | 0.359
>
> Positiveness was highest when using CBF ($\alpha=0.4$), Naturalness was highest when using CBF ($\alpha=0.6$), the number of disallowed tokens was lowest when using CBF ($\alpha=0.8$). The BlockList method is a special case of FUDGE **[4]** with a binary-valued reward function, and the CBF filters outperformed the BlockList method in all metrics.
>
> ---
>
> # Weakness 3
> > **Begins with** "I find the analogy with a c..."
>
> As the reviewer points out, the comparison between a vehicle and LLM was insufficient. In LLM, by filtering the token probability distribution, we make the generated text suitable for the user specification. We have added this explanation to `Appendix A`.
>
> > **...** there is a key difference between the two systems: while vehicle collision avoidance involves direct access to \textit{physical quantities} such as steering angle and brake pedal position,
> intervention-based LLM alignment involves access to the \textit{probability distribution} of generated tokens.
> This difference will be taken into account in the development of our control strategy.
>
> ---
>
> # Weakness 4
> > **Begins with** "It will be good to cite a rele..."
>
> We appreciate the reviewer for introducing a related paper.  We have updated the reference list including the introduced paper **[3]** in the revised paper.
>
> ---
>
> # Reference List
> [1] [Controlled Decoding from Language Models](https://icml.cc/virtual/2024/poster/33639), ICML 2024.
>
> [2] [G-Eval: NLG Evaluation using GPT-4 with Better Human Alignment](https://aclanthology.org/2023.emnlp-main.153/), Proceedings of the 2023 Conference on Empirical Methods in Natural Language Processing.
>
> [3] [Language Model Sentence Completion with a Parser-Driven Rhetorical Control Method](https://aclanthology.org/2024.eacl-short.18/), Proceedings of the 18th Conference of the European Chapter of the Association for Computational Linguistics 2024.
>
> [4] [FUDGE: Controlled Text Generation With Future Discriminators](https://aclanthology.org/2021.naacl-main.276/), Proceedings of the 2021 Conference of the North American Chapter of the Association for Computational Linguistics: Human Language Technologies.

---

### Official Review · Reviewer_qEfD · 2024-11-05

**Soundness:** 2
**Presentation:** 3
**Contribution:** 3
**Rating:** 5
**Confidence:** 4

**Summary:**

This paper tackles the task of LLM alignment from the control engineering perspective. Specifically, a filter is applied to the sequence generation steps of LLMs. The fundamental idea is that if a particular property is maintained at every step, that property should ultimately be evident in the final output.

**Strengths:**

1. Tackling LLM generation from the perspective of control engineering is interesting. In particular, it is novel to  use a hyperparameter $\alpha$ to adjust the tightness of the constraint in autoregressive text generation.
2. The paper is well-written, and the intuition equations are clearly explained.

**Weaknesses:**

1. There is a lack of comparison to existing methods for decoding-time LLM alignments, including:
    1. Mudgal et al., Controlled Decoding from Language Models, 2024,
    2. Huang et al., DeAL: Decoding-time Alignment for Large Language Models, 2024
    3. Yang et al., FUDGE: Controlled Text Generation With Future Discriminators, 2021
2. To some extent, the proposed method can be considered a special case of the methods mentioned in Point 1, where the classifier makes hard decisions rather than giving scores. My educated guess (since the authors have not provided supporting experiments) is that the hard-filtering approach is worse. Here is my reason:
   - The semantics of phrases are often determined by later-generated words/phrases; the hard decisions based on the early phrases may result in unnecessary pruning compared to the soft counterpart (with scorers).
   - Suppose we want to generate a positive movie review; an audience may say, "Despite a slow start, the movie blossomed into a riveting tale that kept me on the edge of my seat."  However, this is most likely not allowed by the proposed approach because the generation is cut off at  the "slow start."

**Questions:**

Is the classifier (RoBERTa) trained on whole sequences or prefixes?

---

> ### Author Response · Authors · 2024-11-23
> **Response to Reviewer qEfD**
>
> We appreciate Reviewer qEfD for introducing related works and presenting ideas for updating our method.  In particular, based on the comment **Weakness 2**, we have extended our CBF-LLM with one-step ahead token prediction to that with multi-step ahead one, which improves the performance of the generation.
>
> ---
>
> # Weakness 1
> We appreciate the reviewer for introducing related works.  We have included a comparison with them in the Introduction and Experiment sections. The details of the comparison in the additional experiment are as follows.
>
> Following the initial submission, we conducted additional generation experiments using the Reddit corpus, which was also utilized for the experiments in the paper "Controlled Decoding from Language Models"**[1]**.  We provided 50 samples each to five CBF filters with different hyperparameters, Blocklist, and NoControl, and performed text generation on 300 samples in total.  We evaluated the generated texts using three methods: evaluating positiveness and naturalness using G-Eval **[2]**, measuring the number of disallowed tokens, and measuring generation time per token. The results are summarized in the following table (the same as `Table 3` at `Appendix F.1` in the revised paper):
>
> Method | \# of Disallowed Tokens per Generation | Generation Time per Token [s] | Naturalness | Positiveness
> -|-|-|-|-
> CBF($\alpha=0.2$) | 575 | 0.118 | 0.605 | 0.585
> CBF($\alpha=0.4$) | 335 | 0.110 | 0.627 | $\bf\underline{0.657}$
> CBF($\alpha=0.6$) | 469 | 0.114 | $\bf\underline{0.679}$ | 0.564
> CBF($\alpha=0.8$) | $\bf\underline{299}$ | $\bf\underline{0.108}$ | 0.660 | 0.535
> Blocklist | 368 | 0.110 | 0.653 | 0.547
> NoControl | 0 | 0.0995 | 0.647 | 0.359
>
> Positiveness was highest when using CBF ($\alpha=0.4$), Naturalness was highest when using CBF ($\alpha=0.6$), the number of disallowed tokens was lowest when using CBF ($\alpha=0.8$), and generation time per token was lowest when using CBF ($\alpha=0.8$). Our CBF filters outperformed the Blocklist method, which is a special case of FUDGE [3] with a binary-valued reward function, in all metrics.
>
> ---
>
> # Weakness 2
> We appreciate for the insightful comment.  The reviewer is correct that decisions based on early phrases can unnecessarily limit the possibilities of the generated text. In response to the comment, we developed a new algorithm, "multi-step ahead CBF-LLM", which controls LLMs by considering multiple steps ahead. We conducted generation experiments using this algorithm and obtained the following results for naturalness, positiveness, and generation time per token (This table is shown as `Table 2` at `Section 4.5` in our revised paper):
>
> Metric | CBF($\alpha=0.8$) | Multi-Step Ahead CBF($H=3,\alpha=0.8$) | Blocklist
> - | - | - | -
> Naturalness | 0.660 | $\bf{\underline{0.695}}$ | 0.653
> Positiveness | 0.535 | $\bf{\underline{0.564}}$ | 0.547
> Generation time per token [s] | 0.108 | 2.31 | 0.0995
>
> Compared to the standard CBF-LLM, the multi-step ahead CBF-LLM demonstrated improved naturalness and positiveness. However, the generation time per token significantly increased, highlighting a trade-off between text quality and computational efficiency.
>
> ---
>
> # Question
> The RoBERTa classifier is trained on whole sequences, so it may not be ideal for evaluating midway sentences.  To note the limitation, we have added the following remark to the `CBF-LLM` section of the revised paper, as follows:
> > Finally, we note the limitation of the L-CF with the RoBERTa model.
> The model is originally trained for evaluating whole sentences, while it is used with midway sentences in this example.
> This may deteriorate the accuracy of the evaluation.
>
> ---
>
> # Reference List
>
> [1] [Controlled Decoding from Language Models](https://icml.cc/virtual/2024/poster/33639), ICML 2024.
>
> [2] [G-Eval: NLG Evaluation using GPT-4 with Better Human Alignment](https://aclanthology.org/2023.emnlp-main.153/), Proceedings of the 2023 Conference on Empirical Methods in Natural Language Processing.
>
> [3] [FUDGE: Controlled Text Generation With Future Discriminators](https://aclanthology.org/2021.naacl-main.276/), Proceedings of the 2021 Conference of the North American Chapter of the Association for Computational Linguistics: Human Language Technologies.

---

> ### Comment · Reviewer_qEfD · 2024-11-25
>
> Thanks for the response. I appreciate the changes in the revised version.
>
> However, I am not sure if the Blocklist baseline is the same as the Blockwise Top-K in [2]. The former is a greedy approach, only allowing scores of considering each token to be better. However, the latter approach samples a few tokens at a time and keeps top-K based on the scorer, which does not seem to apply the same constraint.
>
> Additionally, I wonder if the authors can compare the convergence to the desired distribution between the proposed and FUDGE-like methods.

---

> > ### Author Response · Authors · 2024-11-28
> > **Response to Follow-up Comment from Reviewer qEfD**
> >
> > We sincerely appreciate the additional comment and apologize for our misunderstanding. As the reviewer said, the Blocklist baseline in our paper is not the same as Blockwise Top-K in **[2]**.
> > Blockwise Top-K generates $K$ candidates of $H$-token sequence and then chooses the best one.
> > On the contrary, Blocklist in our paper is considered to be a special case of FUDGE **[3]**.
> >
> > To make clear the Blocklist definition, we added the following description in `Section 4.1`:
> >
> > > The Blocklist filter disallows tokens such that the L-CF $h$ for concatenated text $\mathrm{concat}(x,t)$ indicates a negative value. This method can be seen as a special case of FUDGE, where the FUDGE's reward function $r(x)$ is a binary function that takes $0$ if the text $x$ is undesirable and $1$ if the text $x$ is desirable.
> >
> > Blockwise Top-K in **[2]** is compared with our CBF-LLM with multi-step ahead extension in `Section 4.5`. In the subsection, we evaluated the naturalness by G-Eval and the rate of generated texts that were not positive. The result is summarized in the following table (which is the same as `Table 2` of the revised paper).
> >
> > The result shows the CBF-LLM’s potential in the following scenario (added in `Section 4.5`):
> >
> > > Notably, when the sample size $K$ is small, the Blockwise best-of-$K$ had a relatively high rate of non-positive text generation.
> > Given the practical need to reduce $K$ due to some reason, such as computational efficiency, the multi-step ahead CBF-LLM has potential in scenarios where avoiding undesirable text is guaranteed.
> >
> > We also discussed the topic “the convergence to the desired distribution between the proposed and FUDGE-like methods“ as the remark in `CBF-LLM` section as follows:
> >
> > > FUDGE seeks to align the generated text with a target distribution.
> > Here, we recall the structure of CBF-LLM algorithm.
> > In CBF-LLM, the token probability is modified as the follows:
> > \begin{align}
> >     Q[t] \propto \begin{cases}
> >         P[t] , & \mathrm{concat}(x,t)\text{ satisfies the CBF inequality}, \\\\
> >         0 , & \mathrm{concat}(x,t)\text{ does not satisfy the CBF inequality}.
> >     \end{cases}
> > \end{align}
> > We can see that CBF-LLM aims to constrain the generated text to avoid certain undesirable regions.
> >
> > Thanks to the reviewer’s comments, we made clear the contribution of CBF-LLM. Revisions made in response to the reviewer’s comments are indicated in blue.

---

> > > ### Comment · Reviewer_qEfD · 2024-12-03
> > >
> > > Thanks for the responses and the new version.
> > >
> > > While the current version shows significant improvement, I believe there are still some changes that need to be made for future revisions.
> > >
> > > The main difference between the proposed method and the existing collaborative filtering (CF) method is the use of ranking versus hard filtering. As the authors have noted, one of the advantages of hard filtering compared to the baseline is that it may only require a small value of K to achieve effective control, and I agree with this point.
> > >
> > > I believe the focus of this paper should be more on justifying hard filtering. Specifically, the authors could elaborate on the quality-efficiency trade-off between the top-K and collaborative-based filtering (CBF) methods, examining different settings for K and N. Additionally, I don’t think the top-K method requires a look-ahead mechanism, as this adds unnecessary computation. In other words, if the authors intend to compare the trade-offs, they should also consider the costs associated with using a look-ahead approach.

---

### Official Review · Reviewer_yTZ8 · 2024-11-06

**Soundness:** 1
**Presentation:** 2
**Contribution:** 1
**Rating:** 3
**Confidence:** 4

**Summary:**

This paper deals with safety control in LLM alignment. The authors get inspiration from collision avoidance in control engineering and propose a control barrier function as the safety filter to ensure the user-desirable text. Experiments are conducted on Llama3 using a RoBERTa model as the CBF.

**Strengths:**

- The safety control of LLM outputs is a significant topic.
- It is interesting to see the connection between control engineering and text generation, e.g. the collision avoidance analogy to safety control in text generation.

**Weaknesses:**

- The proposed method is very similar to existing work inference-time constrained decoding such as SafeDecoding[1]. The authors should definitely discuss the line of work.

- Based on my understanding, this work mainly introduces a CBF (specifically sentiment analysis RoBERTa) to measure a metric h(x) during decoding. The details of text generation the authors describe should be the standard greedy decoding process with top-k sampling. Instead of using standard greedy or beam search, they introduce the CBF to measure the toxicity of the generated text and restrict tokens with negative h(x). Then (1) it is so weird and not reliable to use a sentiment classifier to predict the toxicity instead of existing toxicity classifiers (2) even if using a reasonable toxicity classifier, a single token does not necessarily change a text sequence from safe to unsafe state. Existing works usually use guardrails such as Llama-Guard to measure the entire generated text sequence rather than part of the sequence.

- The experiments are not convincing. The only data used in the experiment is just one single sentence. To verify the effectiveness, experiments on existing jailbreak datasets such as AdvBench[2], HarmBench[3] are necessary. To claim its effectiveness on hallucination mitigation, more experiments on related datasets are also necessary rather than using just one sentence.

- It is unclear whether this new decoding algorithm has an impact on the helpfulness of LLMs for other datasets such as MT-Bench.

- The average token generation time also needs to be discussed.

**Questions:**

- Does Section 2.2 describe the standard greedy decoding process (correct me if I am wrong)? I would recommend the authors refer more to the traditional decoding algorithm in the NLP domain, otherwise it would be very confusing to the readers.
- Even though it is interesting to discuss the counterpart and analogy in control engineering, relating it more to existing works in the LLM decoding would make it easier to tell the differences and see if there are real contributions.

[1] Xu, Z., Jiang, F., Niu, L., Jia, J., Lin, B. Y., & Poovendran, R. (2024). Safedecoding: Defending against jailbreak attacks via safety-aware decoding. ACL.

[2] Zou, A., Wang, Z., Carlini, N., Nasr, M., Kolter, J. Z., & Fredrikson, M. (2023). Universal and transferable adversarial attacks on aligned language models. arXiv preprint.

[3] Mazeika, M., Phan, L., Yin, X., Zou, A., Wang, Z., Mu, N., ... & Hendrycks, D. (2024). Harmbench: A standardized evaluation framework for automated red teaming and robust refusal. ICML.

---

> ### Author Response · Authors · 2024-11-23
> **Response to Reviewer yTZ8 (1)**
>
> We appreciate Reviewer yTZ8 for raising many concerns and introducing related works and datasets.  We have conducted additional text generation experiments using a dataset and provided comparisons with existing works.  In addition, we have revised some statements to improve clarify and to highlight our contributions.  The detailed discussion of each comment raised by the reviewer is presented as follows.
>
> # Weakness 1
> We have included a comparison with existing works [1][2][3][4][5] in the Introduction and Experiment sections.
>
> We appreciate the reviewer for introducing SafeDecoding.  But, SafeDecoding, which intervenes only the first token to maintain the safety of the entire sentence, differs from our CBF-LLM, which sequentially intervenes in each token prediction to control the overall text generation process.
>
> Therefore, in the additional experiments, we compared our CBF-LLM with the Blocklist method, which is a special case of FUDGE [5] with a binary-valued reward function, rather than with SafeDecoding. The details of the experiments are described in the response of **Weakness 3**.
>
> # Weakness 2
> As stated by the reviewer, our work mainly introduces CBF to measure a metric $h(x)$ for "safety" during decoding and intervenes the token prediction.
> In `Section 2.2`, the decoding process of an LLM is regarded as a dynamical system, as shown in `eq.(5)`. While this description may initially seem unusual, in particular readers in NLP domain, it can represent standard approaches like the greedy or beam search.  Such a representation of the decoding process as a dynamical system is essential for introducing CBF into controlled decoding.
>
> **(1)**
>
> We appreciate the reviewer for raising concern on the inconsistency within the approach and the control objective.
>
> First, the term "safe control" might be ambiguous in this paper, and we should make clear the meaning in the revised paper.  Let us emphasize that this paper aims to propose a framework for generating text suitable *"for user-specific objectives"*, such as non-toxic (= *"safe"* in common) and positive.  In the main experiment of this paper, we aim to generate positive (not necessarily non-toxic) text using a sentiment analysis model.  In this sense, the approach consists with the objective.  We note that our framework of CBF-LLM is also applicable to generate non-toxic (= *"safe"* in common) text using a toxicity prediction model.
>
> We understood "safety" in a broad sense, such as "positiveness" or "harmlessness".   We have added the comment on the meaning of *"safe control"* in the `Introduction` section of the revised paper as follows:
> > "Safe Control" addressed in this paper reflects a broader interpretation of "safety".
> We mean safe control as the ability to regulate LLM output to produce text that not only adheres to ethical standards but also aligns with user-specific objectives.
>
> **(2)**
>
> Even in the case of a single token intervention, our method has value in that it improves quality compared to the Blocklist approach. However, as the reviewer pointed out, it is essential to consider longer tokens, so we propose a "multi-step ahead algorithm" as an extension of CBF-LLM.  We have included the additional experiment of multi-step ahead CBF-LLM with showing its potentials in `Section 4.5` of the revised paper.  A part of the result is summaried in the following table, which is the same as `Table 2` of `Section 4.5` of the revised paper.
>
> **Table**: Non-Positive Generation Rate / Naturalness
>
> Sample Size $K$ | Multi-Step Ahead CBF-LLM ($H=3,\alpha=0.8$) | Blockwise best-of-$K$ ($H=3$)
> - | - | -
> 2 | $\bf{\underline{0.00}}/\\bf\underline{0.722}$ | $0.30/0.592$
> 4 | $\bf\underline{0.00}/\bf\underline{0.718}$  | $0.02/0.695$
> 5 | $0.00/\bf\underline{0.727}$ | $0.00/0.701$
>
> An explanation for this result is included in `Section 4.5` as follows:
> > Notably, when the sample size $K$ is small, the Blockwise best-of-$K$ had a relatively high rate of non-positive text generation.
> The Blockwise best-of-$K$ method does not disallow the user-undesired outputs, which may lead to user-undesired results.
> In contrast, the multi-step ahead CBF-LLM did not produce any undesirable text for any value of $K$ due to the safety filter.
> Given the practical need to reduce $K$ due to some reason, such as computational efficiency, the multi-step ahead CBF-LLM has potential in scenarios where avoiding undesirable text is guaranteed.
>
> We appreciate for introducing Llama-Guard. While such a guardrail approach aims to *exclude* undesired texts, CBF-LLM aims to *guide* text generation to align with the user-specific objective.

---

> ### Author Response · Authors · 2024-11-23
> **Response to Reviewer yTZ8 (2)**
>
> # Weakness 3
> We conducted additional generation experiments, where 50 samples each to five CBF filters with different hyperparameters, BlockList, and NoControl are utilized for the text generation on 300 sentences in total.  The experiments are based on the Reddit corpus from the paper "Controlled Decoding from Language Models"[2], which the reviewer qEfD introduced to us.
>
> This time, we used a pre-trained foundation model to focus on evaluating the control performance of the baseline decoder. Therefore, we did not use AdvBench and HarmBench, which are benchmarks for response models, as they are not directly applicable to this experiment.  We appreciate for introducing datasets for benchmarks.
>
> To allow you to review the additional experimental results without accessing the revised paper itself, we present part of the experimental results below.
> In the following table (the same as `Table 3` in the revised paper), the results of the text generation experiments are summarized including the evaluation on positiveness and naturalness using G-Eval[6], the number of disallowed tokens, and generation time per token.
>
> Method | \# of Disallowed Tokens per Generation | Generation Time per Token [s] | Naturalness | Positiveness
> -|-|-|-|-
> CBF($\alpha=0.2$) | 575 | 0.118 | 0.605 | 0.585
> CBF($\alpha=0.4$) | 335 | 0.110 | 0.627 | $\bf\underline{0.657}$
> CBF($\alpha=0.6$) | 469 | 0.114 | $\bf\underline{0.679}$ | 0.564
> CBF($\alpha=0.8$) | $\bf\underline{299}$ | $\bf\underline{0.108}$ | 0.660 | 0.535
> Blocklist | 368 | 0.110 | 0.653 | 0.547
> NoControl | 0 | 0.0995 | 0.647 | 0.359
>
> Positiveness was highest when using CBF ($\alpha=0.4$), Naturalness was highest when using CBF ($\alpha=0.6$), the number of disallowed tokens was lowest when using CBF ($\alpha=0.8$). The CBF filters outperformed the Blocklist method, which is a special case of FUDGE [5] with a binary-valued reward function, in all metrics.
>
> # Weakness 4
> As mentioned in the response to **Weakness 3**,  we conducted experiments using a pre-trained foundation model. Helpfulness is a metric targeted at instruct-following models, so we will consider it as future work.
>
> # Weakness 5
> We appreciate this comment.  In the additional experiment, we measured the generation time of each method. The results are shown in the response to **Weakness 2**. Our CBF-LLM experimentally demonstrated the slightly reduced inference time compared to the widely used blocklist approach, $0.108\ \mathrm{sec/token}$ for CBF($\alpha = 0.8$), while $0.110\ \mathrm{sec/token}$ for Blocklist.
>
> # Question 1
> Yes, `Section 2.2` describes the standard decoding process in a modified manner. Although the explanation may be unconventional in the NLP domain, it is necessary for elucidating CBF-LLM. We intentionally framed LLM as a dynamic system to facilitate this explanation.  In the revised paper, we have made clear the connection of the description in `Section 2.2` to traditional decoding algorithm, as follows:
> > The examples of $C$ includes greedy algorithm.
>
> # Question 2
> We agree that a more explicit connection to existing works in LLM decoding would help to clarify the contributions of our research. To address this, we highlight the key differences between collision avoidance in vehicles and LLM. Specifically, while collision avoidance in vehicles involves intervening with physical objects, collision avoidance in LLM involves intervening with probability distributions, as discussed in `Appendix B`, as follows:
> > However, there is a key difference between the two systems: while vehicle collision avoidance involves direct access to *physical quantities* such as steering angle and brake pedal position,
> intervention-based LLM alignment involves access to the *probability distribution* of generated tokens.
> This difference will be taken into account in the development of our control strategy.

---

> > ### Author Response · Authors · 2024-11-23
> > **Response to Reviewer yTZ8 (3)**
> >
> > # Reference List
> > [1] [SafeDecoding: Defending against Jailbreak Attacks via Safety-Aware Decoding](https://aclanthology.org/2024.acl-long.303/), Proceedings of the 62nd Annual Meeting of the Association for Computational Linguistics 2024.
> >
> > [2] [Controlled Decoding from Language Models](https://icml.cc/virtual/2024/poster/33639), ICML 2024.
> >
> > [3] [Language Model Sentence Completion with a Parser-Driven Rhetorical Control Method](https://aclanthology.org/2024.eacl-short.18/), Proceedings of the 18th Conference of the European Chapter of the Association for Computational Linguistics 2024.
> >
> > [4] [DeAL: Decoding-time Alignment for Large Language Models](https://arxiv.org/abs/2402.06147), arXiv 2024.
> >
> > [5] [FUDGE: Controlled Text Generation With Future Discriminators](https://aclanthology.org/2021.naacl-main.276/), Proceedings of the 2021 Conference of the North American Chapter of the Association for Computational Linguistics: Human Language Technologies.
> >
> > [6] [G-Eval: NLG Evaluation using GPT-4 with Better Human Alignment](https://aclanthology.org/2023.emnlp-main.153/), Proceedings of the 2023 Conference on Empirical Methods in Natural Language Processing.

---

### Official Review · Reviewer_jp7H · 2024-11-06

**Soundness:** 2
**Presentation:** 2
**Contribution:** 2
**Rating:** 1
**Confidence:** 4

**Summary:**

The paper studies controllable decoding in LLM generation, e.g., keeping the generated text of a positive sentiment or withing a specific topic. The paper proposes using a classifier that is trained on examples that are specific to the target control criteria. The classifier is used to define a control constraint that is checked repeatedly after generating each token. The probabilities of the candidate tokens that violate the constraint are set to zero. The paper limits the set of candidate tokens to k << vocab-size (e.g., 30) for efficiency. With a single prefix for controlling for the sentiment of the text and another single prefix for controlling for the topic, the paper shows some advantage over not applying any control at all.

**Strengths:**

1. The presented method is learning free and broadly applicable.

**Weaknesses:**

1. The paper does not compare to existing baselines beyond just the naïve blocklist approach. In fact, there are published and peer-reviewed papers that achieve the same goal with the closely related method that the paper does not compare to, e.g., Mudgal et al, "Controlled Decoding from Language Models", ICML 2024.

2. The paper does not present sufficient experimental results. It is just one prefix that is used per each of the two use cases demonstrated. That is not even sufficient for a workshop paper. The paper needs to provide quantitative results that are aggregated across several diverse examples.

3. Related to the weak experiments, the paper also needs to provide human assessment of the produced generations and use that to demonstrate that the presented method truly introduces some value over the blocklist approach. Also, how does the control method impact the generation quality of the model (e.g., fluency, naturalness, general LLM capabilities), what happens if we wanted to control for more than one aspect, what is the role of the data properties (e.g., size) used to train the classifier on the overall quality.

4. Even with limiting the candidate tokens size to the top-30, applying that methods at each decoding step (i.e., token generation) seems expensive. The paper needs to provide some details on that cost.

**Questions:**

Please see my questions under weaknesses.

---

> ### Author Response · Authors · 2024-11-23
> **Response to Reviewer jp7H (1/2)**
>
> We appreciate Reviewer jp7H for insightful and pointed comments, even if they were critical. We have conducted additional text generation experiments using a dataset collected from the Reddit corpus and provided their evaluations.  The detailed discussion of each comment raised by the reviewer is presented as follows.
>
> # Weakness 1
> We have included a comparison with existing works on controlled decoding alignment[1][2][3][4][5] in the `Introduction` and `Experiment` sections.  Specifically, our method is related to the paper [1], which presents the same controlled decoding algorithm as the Blocklist approach in our paper. The results of the comparison are presented in the responses to **Weakness 2** and **Weakness 3**, providing a comprehensive evaluation of our method against established baselines.
>
> # Weakness 2
> Following the initial submission, we conducted additional text generation experiments using the Reddit corpus, which was also utilized for the experiments in the paper "Controlled Decoding from Language Models"[1].  We provided 50 samples each to five CBF filters with different hyperparameters, Blocklist, and NoControl, and performed text generation on 300 samples in total.  We evaluated the generated texts using three methods: evaluating positiveness and naturalness using G-Eval[6], measuring the number of disallowed tokens, and measuring generation time per token. The results are summarized in the following table (the same as `Table 3` in `Appendix F.1` of the revised paper)
>
> Method | \# of Disallowed Tokens per Generation | Generation Time per Token [s] | Naturalness | Positiveness
> -|-|-|-|-
> CBF($\alpha=0.2$) | 575 | 0.118 | 0.605 | 0.585
> CBF($\alpha=0.4$) | 335 | 0.110 | 0.627 | $\underline{0.657}$
> CBF($\alpha=0.6$) | 469 | 0.114 | $\underline{0.679}$ | 0.564
> CBF($\alpha=0.8$) | $\underline{299}$ | $\underline{0.108}$ | 0.660 | 0.535
> BlockList | 368 | 0.110 | 0.653 | 0.547
> NoControl | 0 | 0.0995 | 0.647 | 0.359
>
> Positiveness was highest when using CBF ($\alpha=0.4$), Naturalness was highest when using CBF ($\alpha=0.6$), the number of disallowed tokens was lowest when using CBF ($\alpha=0.8$), and generation time per token was lowest when using CBF ($\alpha=0.8$). Our CBF filters outperformed the Blocklist method, which is
> a special case of FUDGE [5], where the reward function $r(x)$ takes a binary value: $r(x)=0$ if the text $x$ is undesirable and $r(x)=1$ if the text $x$ is desirable,
> in all metrics.  The experiments above with detailed discussions are included in `Section 4.2` and `Appendix F.1` of the revised paper.
>
> # Weakness 3
> We have evaluated the impact of the control method on Naturalness, with results presented in our response to **Weakness 2**. To extend control to multiple aspects, a promising approach is to cascade CBF filters designed independently. However, the performance of the classifier model (denoted as L-CF in the paper) has not been thoroughly investigated with diverse configurations, and its behavior remains unclear.  The remarks on the extensions and potentials of our CBF-LLM have been included in the `Concluding Remark` section of the revised paper as follows:
> > In the other scenario, consider an LLM initially trained or controlled to produce positive text.
> Later, suppose that an additional requirement is added such as ensuring that the generated text is easy for children to comprehend.
> In CBF-LLM, we can independently design a readability CBF filter without modifying the existing positivity CBF filter, allowing the system to meet the updated requirements without having to retrain the entire LLM.
> This approach enables us to easily adapt to changing specifications and requirements.
> In these scenarios, the CBF-LLM approach offers a significant advantage.
>
> # Weakness 4
> As the reviewer pointed out, our CBF-LLM is more computationally expensive compared to the NoControl case. To address the concern about computational cost, we measured the inference time for five different hyperparameter CBF filters, BlockList, and NoControl. Please refer to the response to **Weakness 2** for the results. Our CBF-LLM experimentally demonstrated the slightly reduced inference time compared to the widely used blocklist approach, $0.108\ \mathrm{sec/token}$ for CBF($\alpha = 0.8$), while $0.110\ \mathrm{sec/token}$ for Blocklist.

---

> > ### Author Response · Authors · 2024-11-23
> > **Response to Reviewer jp7H (2/2)**
> >
> > # Reference List
> > ```plaintext
> > [1] Controlled Decoding from Language Models", ICML 2024.
> > [2] Language Model Sentence Completion with a Parser-Driven Rhetorical Control Method, Proceedings of the 18th Conference of the European Chapter of the Association for Computational Linguistics 2024.
> > [3] SafeDecoding: Defending against Jailbreak Attacks via Safety-Aware Decoding, Proceedings of the 62nd Annual Meeting of the Association for Computational Linguistics 2024.
> > [4] DeAL: Decoding-time Alignment for Large Language Models, arXiv 2024.
> > [5] FUDGE: Controlled Text Generation With Future Discriminators, Proceedings of the 2021 Conference of the North American Chapter of the Association for Computational Linguistics: Human Language Technologies.
> > [6] G-Eval: NLG Evaluation using GPT-4 with Better Human Alignment, Proceedings of the 2023 Conference on Empirical Methods in Natural Language Processing.
> > ```

---

### Author Response · Authors · 2024-11-27
**Comment by Authors**

We sincerely appreciate the area chair and the five reviewers for taking the time to review our paper and providing valuable and insightful comments. All review comments have been incorporated into the revised paper to improve the quality. In particular, we have addressed the main concerns raised by the reviewers:

1. the lack of comparisons with previous approaches to controlled decoding methods,
2. the limited number of sample text generations in the experiments, and
3. the insufficient quantitative evaluation of our CBF-LLM.

We have highlighted the revised statements in red and blue in the revised paper. The first revision is indicated in red, and the second revision is indicated in blue.

---

> ### Author Response · Authors · 2024-11-28
> **Comment by Authors**
>
> Thank you for the follow-up response. We followed the suggestions from Reviewer **qEfD** and made minor revisions. In multi-step ahead experiments, we compared with prior research **[1]**. We made clear the contribution of safety assurance of CBF-LLM.
>
> We kindly request your active participation
>
> ---
>
> **[1]** FUDGE: Controlled Text Generation With Future Discriminators, Proceedings of the 2021 Conference of the North American Chapter of the Association for Computational Linguistics: Human Language Technologies.

---

### Meta-Review · Area_Chair_3TPH · 2024-12-19

**Metareview:**

This paper proposes CBF-LLM, a framework that applies control barrier functions from control theory to guide LLM text generation. The core idea is to use a safety filter to intervene the token prediction process to ensure that generated text aligns with desired properties.

While it shows potential to bring control theory to LLM alignment, it has significant gaps in terms of empirical validation. The reviewers raised several major concerns including the limited evaluation and lack of comparison to prior work. Given these fundamental limitations, I recommend rejection for this paper.

**Additional Comments On Reviewer Discussion:**

The reviewers raised several major concerns as summarized above, the authors are responsive and made some improvements, especially with extra experiments. However, some concerns remain serious limitations (efficiency problem, reliance on classifiers), suggesting that the method is not yet ready for publication.

---

### Decision · Program_Chairs · 2025-01-22

Reject